# A dynamic RNA loop in an IRES affects multiple steps of elongation factor-mediated translation initiation

**Marisa D Ruehle[1], Haibo Zhang[2], Ryan M Sheridan[1], Somdeb Mitra[3], Yuanwei Chen[2], Ruben L Gonzalez Jr[3], Barry S Cooperman[2], Jeffrey S Kieft[1,4]\***

[1]Department of Biochemistry and Molecular Genetics, University of Colorado Denver School of Medicine, Aurora, United States; [2]Department of Chemistry, University of Pennsylvania, Pennsylvania, United States; [3]Department of Chemistry, Columbia University, New York, United States; [4]Howard Hughes Medical Institute, University of Colorado Denver School of Medicine, Aurora, United States

**Abstract** Internal ribosome entry sites (IRESs) are powerful model systems to understand how the translation machinery can be manipulated by structured RNAs and for exploring inherent features of ribosome function. The intergenic region (IGR) IRESs from the *Dicistroviridae* family of viruses are structured RNAs that bind directly to the ribosome and initiate translation by co-opting the translation elongation cycle. These IRESs require an RNA pseudoknot that mimics a codon-anticodon interaction and contains a conformationally dynamic loop. We explored the role of this loop and found that both the length and sequence are essential for translation in different types of IGR IRESs and from diverse viruses. We found that loop 3 affects two discrete elongation factor-dependent steps in the IRES initiation mechanism. Our results show how the IRES directs multiple steps after 80S ribosome placement and highlights the often underappreciated significance of discrete conformationally dynamic elements within the context of structured RNAs.

**\*For correspondence:** jeffrey. kieft@ucdenver.edu

**Competing interests:** The authors declare that no competing interests exist.

## Introduction

A vital step in infection by viruses is translation of the viral RNA. Many RNA viruses initiate translation using internal ribosome entry sites (IRESs), which are *cis*-acting RNA elements that recruit the host cell's translation machinery in a cap- and end-independent fashion (*Filbin and Kieft, 2009*; *Doudna and Sarnow, 2007*; *Plank and Kieft, 2012*). Most viral IRESs use a subset of the canonical initiation factor proteins to recruit and position the ribosome, but the intergenic region (IGR) IRESs of the *Dicistroviridae* family of viruses use a more streamlined mechanism (*Figure 1A*). Specifically, the ~200 nucleotide long, compactly folded IRES RNA interacts directly with both ribosomal subunits to assemble 80S ribosomes (*Nishiyama, 2003*; *Costantino and Kieft, 2005*; *Pfingsten et al., 2006*), eliminating the requirement for initiation factors (*Sarnow et al., 2005*; *Jan, 2006*). The IRES binds between the two subunits and, akin to a tRNA, must translocate through the ribosome (*Spahn et al., 2004*; *Schüler et al., 2006*), the only known non-tRNA molecule to do so. In addition, an IGR IRES was recently shown to be able to facilitate translation initiation in live bacteria, although the mechanism in bacteria is very different from the mechanism in eukaryotes (*Colussi et al., 2015*). Current mechanistic models for how the IGR IRESs operate in eukaryotes suggest that after the IGR IRES assembles an 80S ribosome, eukaryotic elongation factor (eEF) 2 catalyzes an initial pseudotranslocation event (translocation without peptide bond formation) which positions the first codon of the open reading frame in the A site (*Figure 1A*) (*Fernández et al., 2014*; *Koh et al., 2014*; *Zhu et al., 2011*). This is followed by eEF1A-catalyzed delivery of the first cognate ac-tRNA to the A site and a

**eLife digest** Many viruses store their genetic information in the form of strands of ribonucleic acid (RNA), which contain building blocks called nucleotides. Once inside an infected cell, the virus hijacks the cellular structures that build proteins (called ribosomes), which forces the cell to start making viral proteins.

Many RNA viruses manipulate the cell's ribosomes using RNA elements called Internal Ribosome Entry Sites, or IRESs. In a family of viruses called *Dicistroviridae*, which infect a number of insects, a section of the IRES RNA binds directly to the ribosome. Proteins called elongation factors then trigger a series of events that lead to the cell starting to make the viral proteins.

By mutating the RNA of many different *Dicistroviridae* viruses that infect a variety of invertebrates, Ruehle et al. have now investigated how a particular loop in the structure of the IRES helps to make cells build the viral proteins. This loop is flexible, and interacts with the ribosome to enable the IRES to move through the ribosome. Mutations that shorten the loop or alter the sequence of nucleotides in the loop prevent the occurrence of two of the steps that need to occur for the cell to make viral proteins. Both of these steps depend on elongation factors. Determining how the entire IRES might change shape as it moves through the ribosome is an important next step, since the ribosome is exquisitely sensitive to the shape and motions of its binding partners.

second eEF2-driven pseudotranslocation event that vacates the A site, allowing delivery of another ac-tRNA, subsequent peptide bond formation, and assumption of the normal translation elongation cycle (*Yamamoto et al., 2007*; *Sasaki and Nakashima, 1999*; *Jan and Sarnow, 2002*; *Pestova, 2003*; *Pestova et al., 2004*). Thus, initiation by this RNA structure-driven process has evolved to use the catalytic action of two GTPase elongation factors. The IGR IRESs have been studied using ribosomes, tRNA, elongation factors, lysate, and cells from sources as diverse as yeast, human, rabbit, shrimp, and wheat germ, often employed in combinations (representative references: *Nishiyama, 2003*; *Costantino and Kieft, 2005*; *Spahn et al., 2004*; *Koh et al., 2014*; *Yamamoto et al., 2007*; *Jan and Sarnow, 2002*; *Pestova, 2003*; *Pestova et al., 2004*; *Cevallos and Sarnow, 2005*; *Wilson et al., 2000*; *Masoumi et al., 2003*; *Thompson et al., 2001*; *Au et al., 2012*; *Costantino et al., 2008*; *Jan et al., 2003*; *Muhs et al., 2015*; *Kamoshita et al., 2009*; *Landry et al., 2009*; *Fukushi et al., 2001*; *Hertz and Thompson, 2011*; *Deniz et al., 2009*; *Jang et al., 2009*; *Pfingsten et al., 2010, 2007*). The mechanism that has emerged is consistent across these systems. This reflects the streamlined IGR IRES mechanism that depends on an RNA structure that manipulates conserved features of the eukaryotic translation machinery. In addition, this feature allows the use of diverse convenient reagents to study the IGR IRESs, a characteristic we took advantage of in this study.

Although IRES structural features that drive formation of the IRES–80S ribosome complex have been mapped, how the IGR IRES co-opts elongation factor function to drive pseudotranslocation through the ribosome is poorly understood. During the canonical elongation cycle tRNA translocation requires specific tRNA–ribosome interactions and conformational states (*Frank et al., 2007*; *Joseph, 2003*; *Schmeing and Ramakrishnan, 2009*; *Frank and Gonzalez, 2010*); it has been proposed that IGR IRESs fulfill these requirements through a strategy that involves both global and local tRNA mimicry (*Costantino et al., 2008*; *Jang et al., 2009*). Globally, the ribosome-bound IGR IRES occupies the spaces normally bound by tRNAs, spans all three tRNA binding sites (*Figure 1—figure supplement 1*) (*Spahn et al., 2004*; *Schüler et al., 2006*; *Fernández et al., 2014*; *Koh et al., 2014*; *Muhs et al., 2015*), interacts with tRNA-binding surfaces on the ribosome, and potentially mimics or induces a hybrid-like state (*Frank et al., 2007*; *Frank and Gonzalez, 2010*; *Moazed and Noller, 1989*). Locally, the IRES mimics tRNA using a pseudoknot-containing domain (pseudoknot I [PKI] in domain III) that structurally mimics the mRNA-tRNA codon–anticodon interaction located just upstream of the translation start site (*Figure 1B*) (*Zhu et al., 2011*; *Costantino et al., 2008*; *Jan et al., 2003*). Previous biochemical and structural studies show that domain III is not needed for initial subunit recruitment and 80S ribosome formation but is essential for establishing the reading frame by docking precisely in the ribosome's decoding groove (*Nishiyama, 2003*; *Costantino and Kieft, 2005*; *Jan and Sarnow, 2002*). However, domain III has features that suggest additional roles.

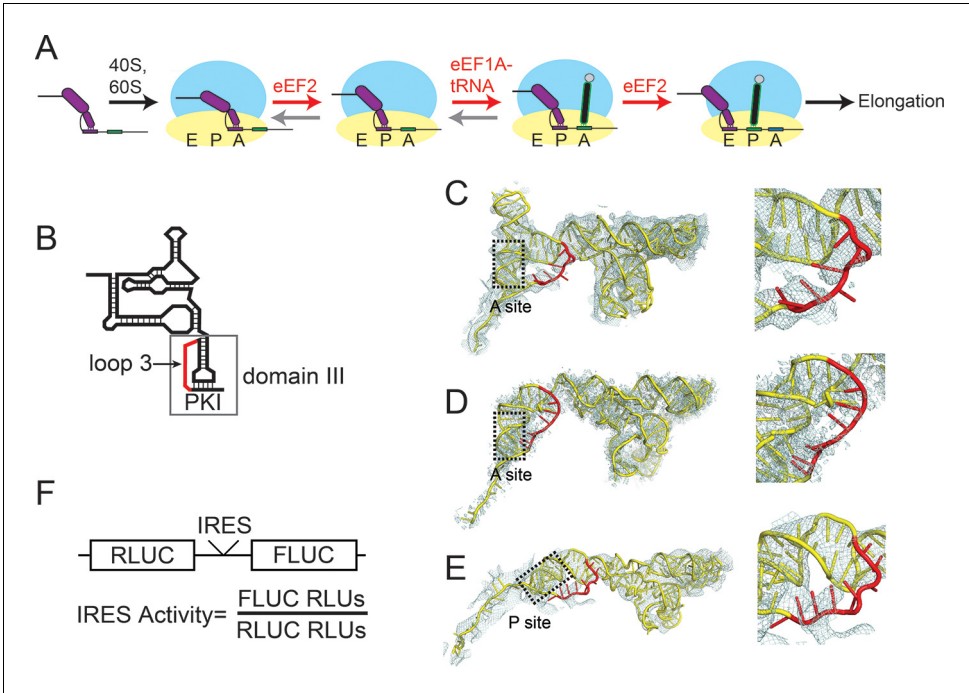

**Figure 1.** Intergenic region (IGR) internal ribosome entry site (IRES) mechanism and loop 3. (**A**) Schematic of the IGR IRES initiation factor-independent translation initiation mechanism. The IGR IRESs occupy the same binding sites as tRNAs in the ribosome. Elongation factor-catalyzed steps are shown in red type and arrows, and proposed reverse reactions are shown with gray arrows. (**B**) Secondary structure cartoon of an IGR IRES with domain III boxed and loop 3 in red. PKI in the figure denotes the pseudoknot base pairs that mimic the codon–anticodon interaction. (**C**) Cryo-electron microscopy (cryo-EM) reconstruction of the Taura Syndrome Virus (TSV) IGR IRES bound to *Saccharomyces cerevisiae* 80S ribosomes (**Koh et al., 2014**). The TSV IRES RNA model is shown in yellow, with loop 3 in red. Density within 8 Å of the IRES model is shown, at a threshold of 2.5. To the right is a close-up view of loop 3. (**D**) Same as panel C, but of a Cricket Paralysis Virus (CrPV) IGR IRES bound to *Kluyveromyces lactis* 80S ribosomes (**Fernández et al., 2014**). Density within 4 Å of the IRES model is shown, at a threshold of 2.5. (**E**) Same as panel C, but of a CrPV IGR IRES bound to *Oryctolagus cuniculus* 80S ribosomes with eukaryotic release factor 1 (eRF1) bound (**Muhs et al., 2015**). Density within 5 Å of the IRES model is shown, at a threshold of 3.0. (**F**) Diagram of the dual luciferase (LUC) reporter RNA used in all in vitro translation assays. IRES activity is determined as a ratio of Firefly LUC activity to Renilla LUC activity.

The following figure supplements are available for Figure 1:

**Figure supplement 1.** IGR IRES location in viral RNA, and alignment and structure of domain III.

**Figure supplement 2.** Loop 3 composition and length in diverse IGR IRESs.

Specifically, x-ray crystal structures of domain III in both the unbound form and bound to ribosomes (**Zhu et al., 2011**; **Costantino et al., 2008**), and chemical probing experiments (**Jan and Sarnow, 2002**; **Pfingsten et al., 2010, 2007**), revealed that the single-stranded loop of RNA ('loop 3') that links the anticodon-like hairpin to the mRNA-like sequence is conformationally dynamic (**Figure 1B**). Mutation or elimination of some bases in loop 3 affects IRES function, purportedly by impairing ribosome positioning, although other effects are possible (**Au et al., 2012**). Cryo-electron microscopy reconstructions provide structural models for loop 3 but the electron density corresponding to this loop is generally weaker than in other parts of the IRES, not continuous, or of low resolution (**Figure 1C–E**) (**Schüler et al., 2006**; **Fernández et al., 2014**; **Koh et al., 2014**; **Muhs et al., 2015**), again suggesting conformational dynamics or structural heterogeneity. These observations are surprising, as domain III comprises an H-type pseudoknot in which the analogous loop usually forms a stable structure (**Staple and Butcher, 2005**; **Aalberts, 2005**; **Westhof and Jaeger, 1992**). Comparing the sequences of IGR IRESs from different species reveals conservation in terms of the length

range and base composition, in particular a high adenosine content (*Figure 1—figure supplements 1, 2*). Adenosine residues in pseudoknot loops often form stable tertiary contacts that are not observed in domain III (*Staple and Butcher, 2005*; *Aalberts, 2005*). These features, combined with our previous work showing that conformationally dynamic structural elements in the IGR IRES can play important roles in IRES function (*Pfingsten et al., 2010*), led us to analyze the mechanistic role of loop 3, focusing on the poorly characterized events following 80S ribosome recruitment.

We discovered that conformationally dynamic loop 3 operates within the context of the highly structured IRES RNA to influence the activity of elongation factors co-opted to drive initiation. We found that both the length and sequence of loop 3 are essential for efficient translation initiation in IGR IRESs from diverse members of the *Dicistroviridae* family. Using the IGR IRES from Cricket Paralysis Virus (CrPV), we demonstrate that loop 3 affects multiple eEF-directed steps, including both pseudotranslocation events. Our findings provide an example of how RNAs can use dynamic regions within the context of a globally stable structure to facilitate function. Because loop 3 is unlikely to interact directly with elongation factors and translocation is a process that depends on ribosome conformational dynamics, our data also suggest a hypothesis in which loop 3 affects ribosome conformations to assist in non-canonical translocation.

## Results

### Loop 3 is important for translation in both IGR IRES classes

We assessed the functional importance of loop 3 in IGR IRES-driven translation using a dual luciferase (LUC) reporter construct in rabbit reticulocyte lysate (RRL) (*Figure 1F*). RRL was chosen as it has proven to be a consistent system for examining the activity of most IGR IRESs. First, we measured the relative translation initiation efficiencies of several IGR IRES RNAs in RRL (*Figure 2A*). Based on this, we chose representative IRESs with differing activities, including Class I and II IGR IRESs (from the *Cripa-* and *Apara-virus* subfamilies), to study the role of loop 3. We made several mutants (*Table 1*): (1) we shortened loop 3 by three nucleotides, reasoning this would reduce flexibility that may be important for function (△3 mutants); (2) noting the loops' high adenosine content, we replaced several adenosines with guanosines (G-rich mutants); (3) because sequence alignment from various IRESs suggested the presence of conserved bases in loop 3 (*Figure 1—figure supplement 1B*) (*Au et al., 2012*), we replaced a single conserved adenosine with a guanosine in the highly active Israeli Acute Paralysis Virus (IAPV) IRES. These mutants are similar to those studied by *Au et al., (2012)*, but are more aggressive in the sense that we deleted more nucleotides (three) and substituted more bases (three). Each mutation had a substantial impact on IRES activity (*Figure 2B,C*). Thus, loop 3 plays a functional role in IGR IRES activity, and this role is shared by diverse members of both IRES classes.

Having established the conserved functional importance of loop 3, we selected the CrPV IGR IRES as a model IRES for additional exploration because it has been widely studied biochemically and structurally, and also because it has the aforementioned characteristic of displaying a consistent mechanism of action when studied with a variety of reagents from diverse species. Several more mutants were designed to assess the importance of loop 3 (*Figure 2D,E*). Shortening loop 3 in the CrPV IGR IRES by just one nucleotide (△1) had a small effect on function while deleting two nucleotides (△2) caused a significant loss of activity; this agrees with previous results (*Au et al., 2012*). The △3 mutant's activity is even more substantially reduced, matching the activity of the negative control PKI/III knockout mutant (*Jan and Sarnow, 2002*; *Costantino et al., 2008*). Likewise, CrPV IRES mutants analogous to the aforementioned G-rich mutants and another mutant in which three conserved bases were mutated (GGC mutant) were substantially decreased in their abilities to initiate translation. Because these differences in measured IRES activity could be due to different amounts of input reporter mRNA or rates of mRNA degradation, we controlled for this in two ways. First, the presence of the upstream Renilla LUC (not under IRES control) provides an internal normalization control for small differences in the amount of RNA in the reaction. Second, we measured the rates of degradation of all reporter mRNAs in the RRL translation reaction, finding that all were equal (*Figure 2—figure supplement 1*). These data indicate that both loop 3 base composition and length are important for CrPV IGR IRES function, and the mutants now provide a set of tools for querying the specific mechanistic role of loop 3.

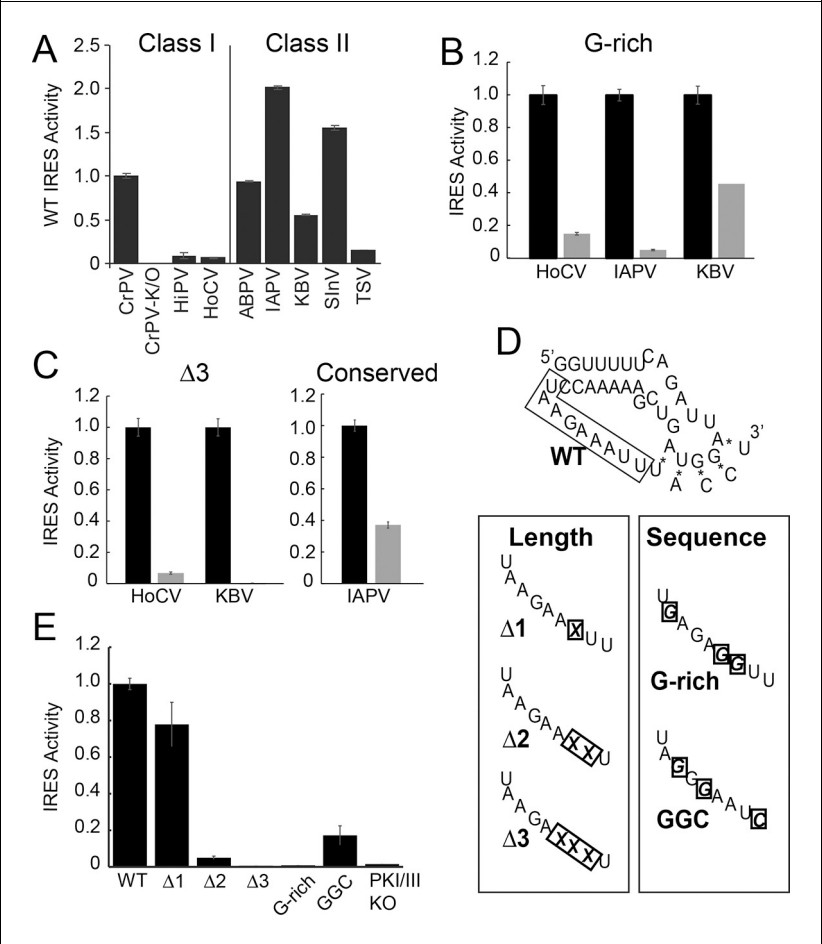

**Figure 2.** Function of diverse wild type (WT) and loop 3 mutant intergenic region (IGR) internal ribosome entry site (IRESs) in rabbit reticulocyte lysate (RRL). (A) Activity of different WT IGR IRESs. Mutant Cricket Paralysis Virus (CrPV)-K/O has pseudoknots III and I disrupted and is the negative control (*Jan and Sarnow, 2002*; *Costantino et al., 2008*). (B and C) Function of WT IRESs (black bars) and loop 3 mutants (gray bars). WT levels are normalized to 1 for each IRES. (D) Diagrams of CrPV IGR IRES domain III mutants. Mutations are boxed and X indicates deletion of a nucleotide. (E) Activity of CrPV loop 3 mutants in RRL. Error bars represent standard error of the mean over at least three biological replicates.

The following figure supplements are available for Figure 2:

**Figure supplement 1.** Degradation of input reporter mRNA in RRL.

## Loop 3 affects an early step in the initiation mechanism, after 80S assembly

Numerous direct ribosome binding studies have shown that domain III can be completely removed or the PKI interaction abrogated without decreasing the IRES's affinity for the ribosome (*Nishiyama, 2003*; *Costantino and Kieft, 2005*; *Jan and Sarnow, 2002*). This suggests that the effects we observe when loop 3 is mutated are not due to alterations in 80S ribosome binding, but rather in events downstream of initial ribosome recruitment. To test this prediction, we used radiolabeled IRES RNAs in RRL to generate IRES–ribosome complexes and resolved them by ultracentrifugation through a sucrose gradient, using an antibiotic to halt the complexes after initial formation (*Figure 3—figure supplement 1A*). All loop 3 mutants robustly assemble 80S–ribosome complexes in RRL. Although there is some variability in the amount of 80S complexes produced in this assay, the amounts do not correlate with the translation activity levels. As a second test for ribosome binding, we measured the approximate on- and off-rates of two mutant IRESs with purified ribosomes from

**Table 1.** Activity of IGR IRESs in RRL and mutations tested.

| Virus | WT activity | Loop 3 mutants tested* | | |
|---|---|---|---|---|
| **Class I** | | G-rich | △3 | Conserved |
| CrPV | ++++ | | | |
| HiPV | + | | | |
| HoCV | + | UUAG***GG***G***CC***G | UUAGA - - - CA | |
| PSIV | + | | | |
| **Class II** | | | | |
| ABPV | ++++ | | | |
| IAPV | +++++ | GA***GG***U***G***CCA | | G***G***AAUACCA |
| KBV | ++ | GAA***GU***G***CC***G | GAAAUA - - - | |
| SInV | ++++ | | | |
| TSV | + | | | |

*Site of mutation is shown in bold italics. Site of deletion is shown as a dash. ABPV, Acute Bee Paralysis Virus; CrPV, Cricket Paralysis Virus (CrPV); HiPV, Himetobi P Virus; HoCV, *Homalodisca coagulata* Virus; IAPV, Israeli Acute Paralysis Virus; IGR, intergenic region; IRES, internal ribosome entry site; KBV, Kashmir Bee Virus; PSIV, *Plautia stali* Intestinal Virus; RRL, rabbit reticulocyte lysate; SInV, *Solenopsis invicta* Virus-1; TSV, Taura Syndrome Virus.

yeast and shrimp sources using filter binding (*Figure 3—figure supplement 1B*). We chose yeast and shrimp ribosomes to complement the RRL and also to test a different source of ribosomes to enable their use in subsequent assays. The measured rates are the same for wild type (WT) and mutant IRES RNAs. Taken together, these data are consistent with the conclusion that the functional effects of mutating loop 3 cannot be accounted for by defects in initial ribosome association with the IRES.

To explore events after initial ribosome binding, we used toeprinting assays to determine if the mutant IRESs are properly positioned within the decoding groove of 80S ribosomes and if they are competent to pseudotranslocate. We chose RRL to match the translation activity assays. Since rabbit and yeast ribosomes produce an identical pretranslocation (PRE) toeprint at the +14/15 position (*Figure 3—figure supplement 2*), we used yeast 80S ribosomes as a marker for the initial IRES location in the 'pretranslocated' state (*Figure 3A* lanes 2 and 18). Toeprinting of the WT CrPV IGR IRES in RRL supplemented with the elongation inhibitor cycloheximide (CHX) reveals that the IRES translocates twice (+20/21 toeprint, *Figure 3A* lanes 3 and 19) as previously observed (*Wilson et al., 2000*). Without CHX no strong toeprints are seen, indicating that the antibiotic traps IRES–ribosome complexes that can be observed in this assay.

Like WT, all length mutants (△1, △2, △3) have a pretranslocated toeprint at +14/15 when bound to pure yeast ribosomal subunits, indicating these IRESs are correctly positioned within the decoding groove of 80S complexes (*Figure 3A* lanes 6, 10, 14). However, in RRL the loop 3 length mutants retain the +14/15 toeprint both with and without CHX to a degree that is roughly inversely correlated with their translation activities, showing that pseudotranslocation is inhibited (lanes 7, 8, 11, 12, 15, 16). A mutation that abrogates codon–anticodon base pairing in PKI does not generate a PRE toeprint at all (*Jan and Sarnow, 2002*); the fact that each mutant IRES still exhibited a PRE toeprint indicates that the mutations tested here probably do not disrupt pseudoknot formation. Furthermore, the +20/21 toeprint is decreased in the △2 mutant and is completely missing in the △3 mutant. The decreases in the +20/21 toeprint are accompanied by an increase in the pretranslocated toeprint, consistent with a decrease in the ability to undergo the first two rounds of pseudotranslocation.

Our experience with the toeprinting method leads us to take great care not to use toeprinting as a quantitative assay of the amount of ribosome binding, given the nature of the assay (not at equilibrium conditions, detected indirectly by reverse transcription, etc.). In general, we conservatively use toeprinting as a robust way to assess the position of ribosomes that are bound, and their movements. After normalization of the signal and with analysis of many replicates, we determined the change in toeprint band intensities at the +14/15 and +20/21 positions to get a semi-quantitative

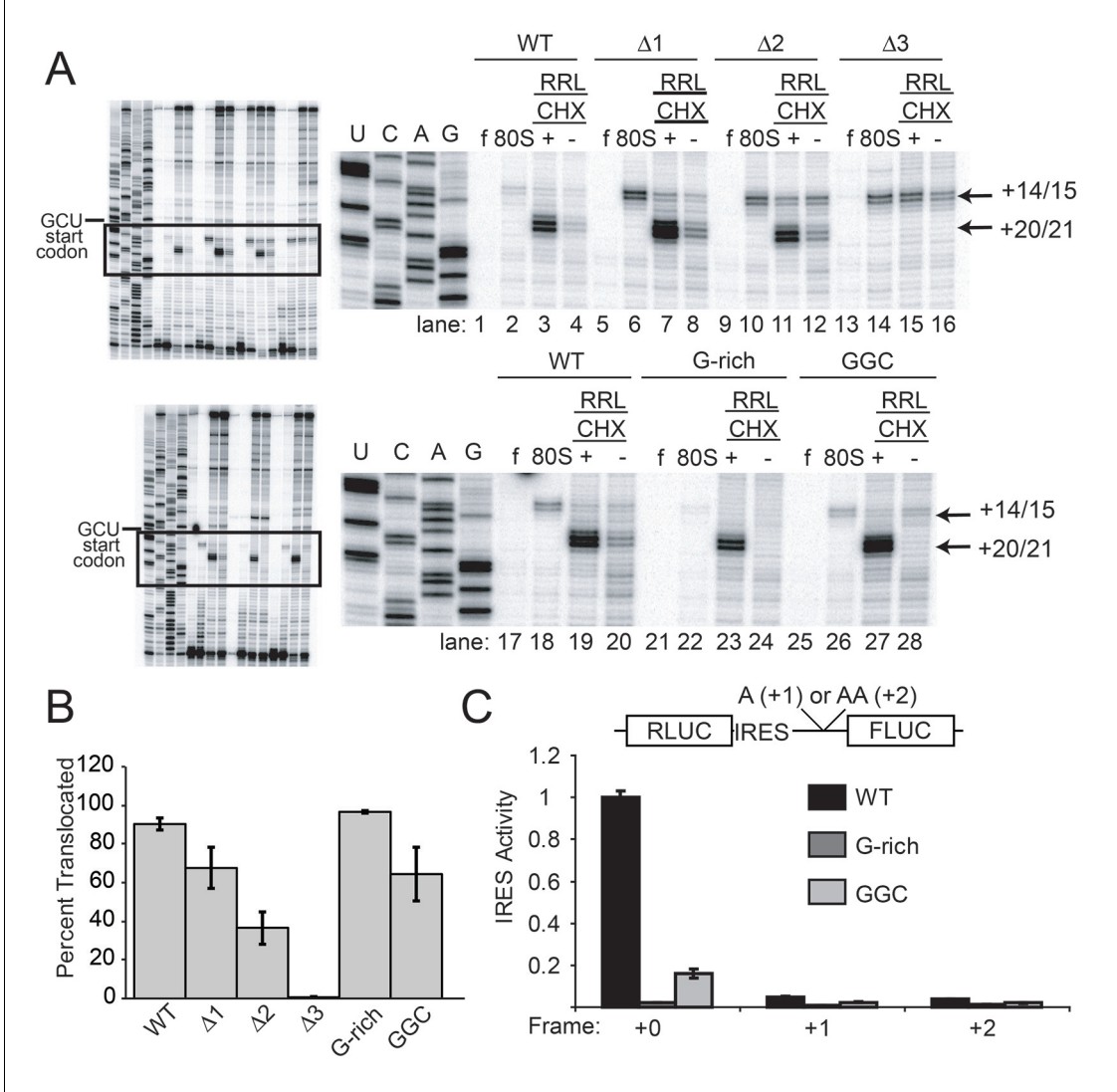

**Figure 3.** Ribosome docking, translocation, and reading frame maintenance. (**A**) Toeprinting analysis of Cricket Paralysis Virus (CrPV) wild-type (WT) internal ribosome entry site (IRES) and loop 3 mutants in the free (f) and yeast 80S ribosome-bound (80S) forms, and in rabbit reticulocyte lysate (RRL) with or without 3 mg/ml cycloheximide (+/- CHX). The +14/15 toeprint indicates the position of the edge of the pretranslocation ribosome, and the +20/21 toeprint shows the position of the edge of the 2x translocated ribosome. Gels are representative of at least six independent experiments. (**B**) Quantification of translocated toeprint bands (+20/21/((+14/15)+(+20/21))) in RRL+CHX (n = 6–9), error bars represent standard error of the mean. (**C**) In vitro translation assay of dual luciferase reporters with +0 (normal), +1, or +2 reading frames. Error bars represent standard error of the mean of three independent experiments.

The following figure supplements are available for Figure 3:

**Figure supplement 1.** IGR IRES loop 3 mutants bind the 80S ribosomes.

**Figure supplement 2.** Toeprinting of WT CrPV IGR IRES with purified 40S subunits and 40S + 60S (80S) ribosomes from two sources.

measure of the percent of ribosomes that successfully perform two pseudotranslocations (*Figure 3B*). In contrast to the measurements of 80S ribosome binding, these data show that shortening loop 3 inhibits the first two steps of pseudotranslocation in a way that correlates very well with the measured translation activity (*Figure 3B and 2E*). Like the length mutants, the G-rich and GGC sequence mutants also form 80S complexes that are properly positioned at the +14/15 location (*Figure 3A* lanes 22 and 26). However, these sequence mutants match WT's ability to generate a

strong +20/21 band (lanes 23 and 27), suggesting they assemble 80S complexes that can translocate (*Figure 3B*). To verify the results with CHX, we performed toeprinting in RRL with the translocation inhibitor hygromycin B, which binds the ribosome in a different location and has a different mechanism of action than CHX (*Borovinskaya et al., 2008*; *Wilson, 2014*) (*Figure 4A*). The WT, G-rich, and GGC mutants pseudotranslocate once (+17/18 toeprint), but the length mutants show a decreased ability to execute the first pseudotranslocation event. Taken together, these data indicate that mechanistic steps affected by loop 3 include the first pseuodotranslocation events after 80S ribosome association.

To identify the step at which the G-rich and GGC mutants are inhibited, we adapted the toeprinting assay to examine their effect on rounds of translocation after the two allowed by CHX. Dilute hygromycin B was added to RRL after addition of IRES RNA (in the experiments described above, RRL was pretreated with high concentrations of hygromycin B or CHX). By altering the concentration of hygromycin B and the time when it was added, we were able to empirically capture the positions of ribosomes after they had loaded and started elongation. WT IRES toeprinting shows four–five rounds of translocation (*Figure 4B*, lane 2). As expected, △1 behaved similarity to WT while the △2 and △3 mutants did not proceed past the initial binding location (lanes 6 and 8). Surprisingly, the sequence mutants displayed toeprinting patterns similar to WT (lanes 10 and 12), although the bands generated from the first few rounds of translocation are less intense, assessed after careful normalization (*Figure 4B*, right). Thus, the G-rich and GGC mutants can translocate at least four–five times in RRL, and the source of their reduced translation initiation activity must be more subtle than a complete failure to translocate. Although all of the mutants showed defects in translation initiation (*Figure 2E*), the toeprinting data indicate that the reasons differ between the length and sequence mutants. The G-rich and GGC mutants do not completely block translocation while the length mutants do, indicating loop 3 has two independent roles in IGR IRES- driven translation initiation.

## Loop 3 mutants do not alter the reading frame

The ability of the G-rich and GGC mutants to translocate in the toeprinting assays suggests they disrupt a different process than do the length mutants. Domain III is essential for establishing the proper reading frame, so the mutations could induce the ribosome to initiate out-of-frame. To test this, we measured translation in RRL using dual LUC constructs with one or two additional nucleotides inserted immediately before the AUG of the firefly LUC open reading frame (+1 and +2 frames), which could rescue out-of-frame initiation (*Figure 3C*). Neither alternate frame rescues IRES activity in the G-rich or GGC loop 3 mutants, indicating the G-rich and GGC mutants do not induce out-of-frame initiation.

## Peptide synthesis is affected by loop 3

If the G-rich and GGC mutants initiate in-frame and can translocate at least four times as indicated by the toeprinting assay, why is their translation activity decreased? It is unlikely that loop 3 acts after the IRES no longer interacts with the ribosome, the presumed situation after four translocation events. Alternatively, decreased toeprint band intensity in these mutants (*Figure 4B* lanes 10 and 12) suggested there could be subtle changes in kinetics of the translocation events. Because toeprinting is not an ideal assay to examine this, we directly explored differences in the rate of peptide synthesis between the WT and the sequence mutants in an in vitro reconstituted translation system by quench-flow (diagrammed in *Figure 5—figure supplement 1*). For these experiments, we used ribosomes from yeast or shrimp eggs, reflecting "one of" the *Dicistroviridae*'s natural arthropod hosts, elongation factors from yeast, and tRNAs of either bacterial or yeast origin. As mentioned above, the use of convenient and high-activity heterologous systems is prevalent in IGR IRES research, and is justified because IGR IRESs appear to function identically in all tested eukaryotic systems. Where appropriate, we indicate the source of each component of the reconstituted system.

Because toeprinting suggested at least four rounds of translocation on the G-rich and GGC mutants in RRL, we first assayed the rate of conversion of tripeptide to tetrapeptide on shrimp ribosomes with the coding sequence for the peptide "Phenylalanine-Valine-Lysine-Methionine" (FVKM) placed downstream of the IRES. Compared to WT, both the G-rich and GGC mutants displayed substantially decreased abilities to convert tripeptide to tetrapeptide, at levels that reflected their relative translation activities (*Figure 5A*). These data suggest that the loss of translation activity in the

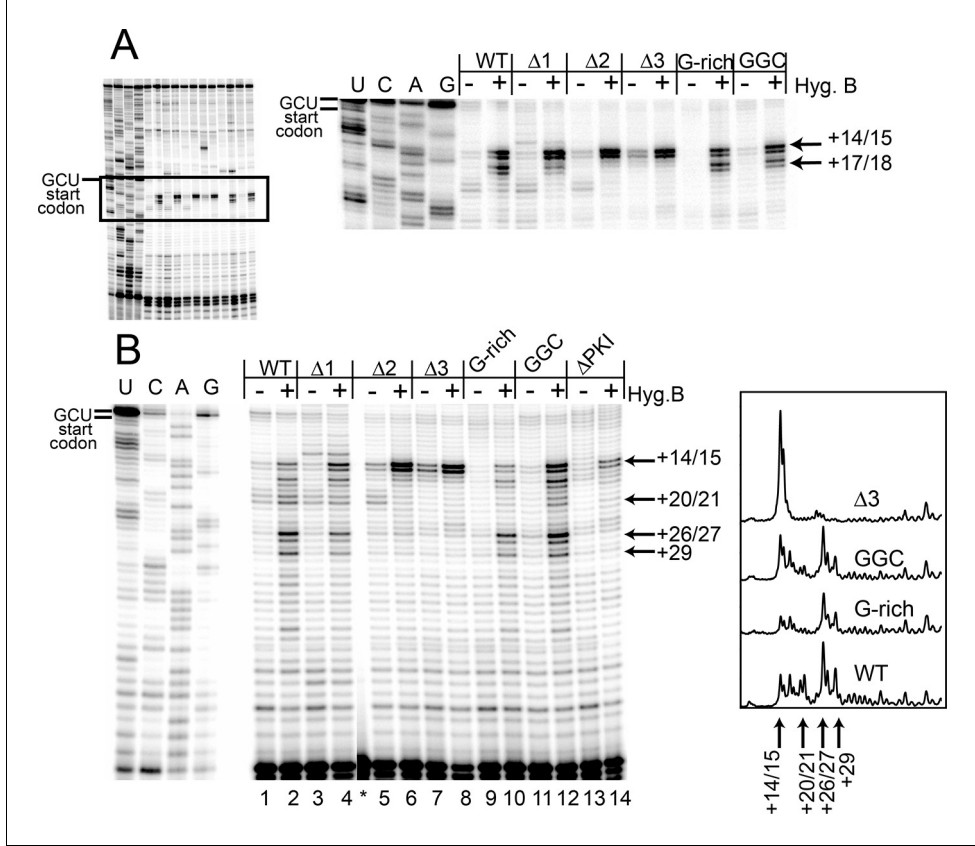

**Figure 4.** Toeprinting with hygromycin B. (**A**) Toeprinting analysis in rabbit reticulocyte lysate (RRL) without or with 0.66 mg/mL hygromycin B (-/+). (**B**) Toeprinting analysis in RRL without or with 3.33 µg/mL hygromycin B (-/+) added after 1 min of incubation of the internal ribosome entry site (IRES) in lysate. Normalized traces of the wild type (WT), △3, G-rich, and GGC IRES RNAs in RRL+ hygromycin B are shown at right. Image is from a single gel, asterisk indicates where two irrelevant lanes were removed.

The following figure supplements are available for Figure 4:

**Figure supplement 1.** RNase T1 probing (single-stranded G bases) of unbound WT, △3, and G-rich Cricket Paralysis Virus (CrPV) intergenic region (IGR) IRES RNAs containing only domain III.

loop 3 sequence mutants is imparted by at least one defective elongation step at or preceding tetra-peptide formation.

## Loop 3 regulates ac-tRNA binding to IRES–ribosome complexes

The decreased peptide synthesis described above could result from inhibition of any step preceding tetrapeptide formation, including binding of the first ac-tRNA to the IRES–80S ribosome complex. To measure the efficiency of this step, we delivered [3H]Phe-tRNA$^{Phe}$ to WT and mutant 80S–IRES (coding for FVKM) shrimp ribosome complexes in the presence of eEF1A-GTP (which forms a ternary complex, TC, with ac-tRNA) and eEF2-GTP and collected these complexes by ultracentrifugation through a sucrose cushion (diagrammed in *Figure 5—figure supplement 1*). As expected, ac-tRNA delivered by eEF1A and translocated to the P site by eEF2 bound stably enough to survive this purification, whereas A-site associated ac-tRNA did not (*Figure 5—figure supplement 2*) (*Yamamoto et al., 2007*). Furthermore, ac-tRNA delivery and binding to the P site depended on a cognate codon–tRNA anticodon interaction (*Figure 5—figure supplement 2*). This latter control is important as it shows that the delivery and binding event we observe in this experiment depends on the presence of the IRES and the placement of the correct codon directly downstream of the IRES within the A site. Therefore, this assay measures the efficiency of completion of all three eEF-

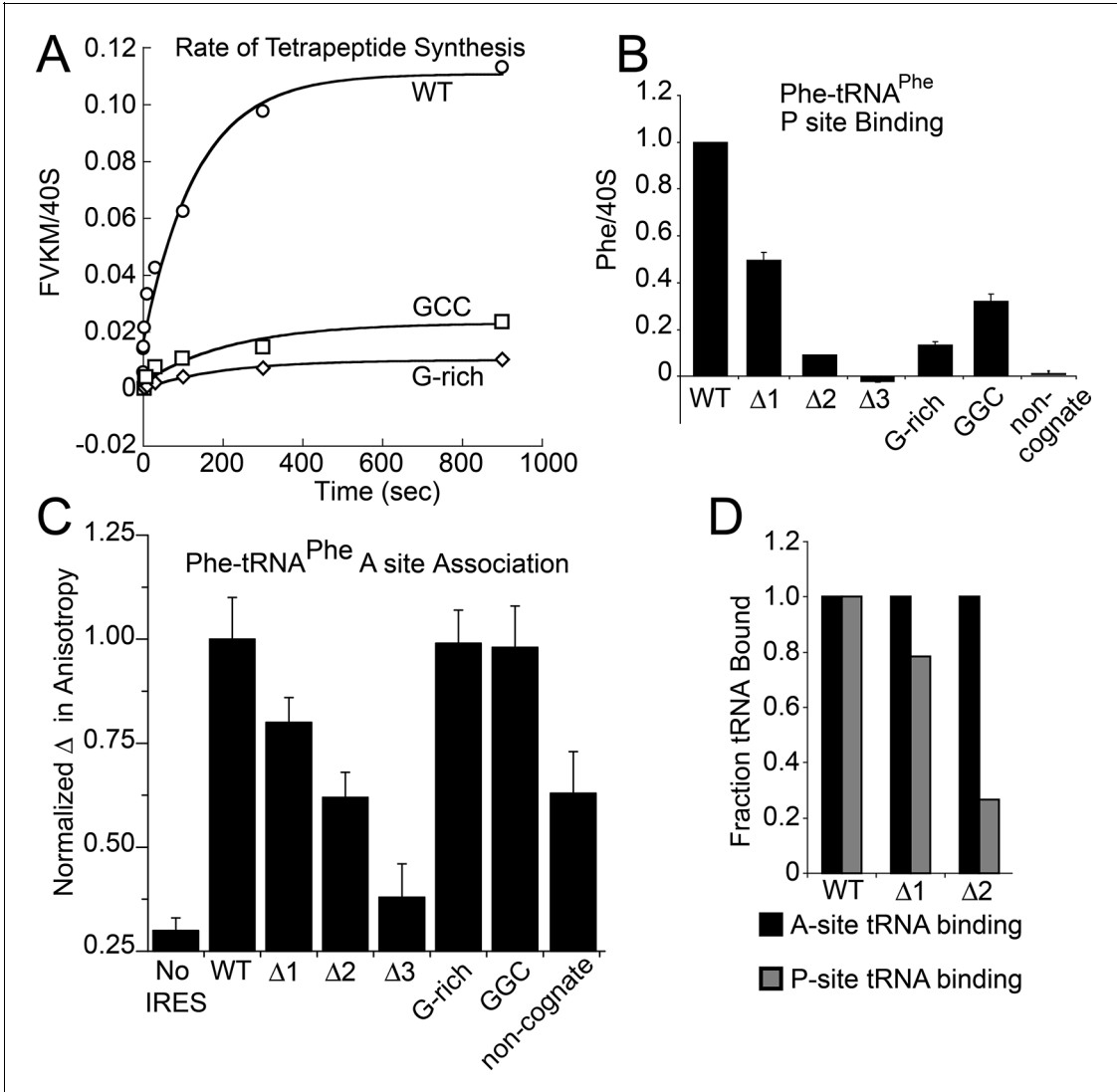

**Figure 5.** Characterization of early steps in intergenic region (IGR) internal ribosome entry site (IRES) initiation in a reconstituted translation system, using purified shrimp ribosomes and yeast elongation factors. (A) Time course of tetrapeptide formation from tripeptide. Data are representative of two independent experiments. (B) [³H]Phe-tRNA^Phe binding to the P site in the presence of eukaryotic elongation factor 2 (eEF2). Triplicate reads were averaged and normalized to set wild type (WT) equal to 1. (C) Anisotropy measurements of Phe-tRNA^Phe(prf) binding to IRES–80S ribosome complexes. For each set of experiments performed, a determination was made of the anisotropy difference (△) between free ternary complex (TC) and TC added to the WT IRES–80S complex, and differences between TC added to other complexes and free TC were normalized to this value. Error bars represent one standard error from the mean of two–four replicates. (D) Translocation efficiency of ac-tRNA from the A to the P site in the △1 and △2 mutants. Data were normalized to set the anisotropy-based A site binding levels (data from C) to 1, and those factors were applied to the cosedimentation-based P site binding levels (data from B).

The following figure supplements are available for Figure 5:

**Figure supplement 1.** Schematic overviews of experiments performed in the reconstituted system.

**Figure supplement 2.** Codon- and factor-dependent tRNA binding to IRES–80S complexes.

**Figure supplement 3.** Normalized anisotropy data.

**Figure supplement 4.** Raw anisotropy data of controls.

dependent steps (*Figure 1A*). As expected, stable [³H]Phe-tRNA^Phe binding was observed with WT IRES with eEF2 (*Figure 5B*), consistent with previous reports (*Yamamoto et al., 2007*). When mutants △1, △2, and △3 were assayed, they showed a progressive decrease in bound [³H]Phe-tRNA^Phe. Interestingly, the G-rich and GGC mutants also showed decreased P-site ac-tRNA association with IRES–80S ribosome complexes at levels that mirror their relative translation activities. Therefore, mutations to loop 3 length and base composition cause decreased association of the first ac-tRNA in the P site.

Because eEF2-GTP was included in the above experiment, we could not distinguish whether decreased ac-tRNA association in the P site resulted from reduced eEF2-driven pseudotranslocation of domain III from the A site to the P site, subsequent ac-tRNA delivery to the A site, or the second pseudotranslocation that moves ac-tRNA from the A site to the P site. To discriminate between these possibilities, we employed a fluorescence anisotropy experiment in which proflavin-labeled Phe-tRNA^Phe [Phe-tRNA^Phe(prf)] TC was delivered to WT and mutant IGR IRES–80S ribosome complexes (shrimp ribosomes) in the absence of eEF2 (diagrammed in *Figure 5—figure supplement 1*). The measured anisotropy of unbound Phe-tRNA^Phe(prf) was 0.205 +/- 0.002 (*Figure 5—figure supplement 4*). As expected, addition of eEF1A-GTP to the ac-tRNA resulted in an increase in measured anisotropy to 0.210 +/- 0.003, consistent with formation of the eEF1A+GTP+Phe-tRNA^Phe(prf) TC. Addition of empty 80S ribosomes (lacking an mRNA or IRES, indicated as 'no IRES') resulted in only a slight increase in change in anisotropy relative to the TC alone (*Figure 5C*). However, when a complex of CrPV IGR IRES bound to 80S ribosomes was added to the TC, we observed a much larger increase in anisotropy, to 0.272 +/- 0.006. This change in anisotropy between TC alone and in the presence of 80S ribosomes+IRES (0.061 +/- 0.003) is consistent with delivery of ac-tRNA to the A site of the IRES–80S ribosome complex by the TC.

To verify that IRES-dependent delivery of tRNA was specific for the first codon following the IRES, we delivered ac-tRNA to an IRES–80S ribosome complex in which the UUC codon for tRNA^Phe was replaced by the non-cognate GCU codon ('non-cognate', *Figure 5C*). This resulted in a smaller increase in anisotropy compared to the IRES with a cognate Phe codon, but larger than the 'no IRES' control. Importantly, the observation that eEF2-independent ac-tRNA binding to the ribosome requires a cognate codon is consistent with the idea that the first codon enters the A site and is queried by the ac-tRNA anticodon. This supports the idea that domain III can spontaneously move to the P site to some degree, perhaps akin to the observed ability of tRNAs to undergo slow spontaneous translocation on bacterial ribosomes (*Gavrilova et al., 1976*; *Gavrilova and Spirin, 1971*; *Pestka, 1969*; *Southworth et al., 2002*; *Fredrick and Noller, 2003*; *Moore, 2012*; *Robertson and Wintermeyer, 1987*; *Semenkov et al., 1992*). The nature of the ac-tRNA's association with the ribosome likely differs depending on whether an IRES RNA with a non-cognate or cognate codon is present; the former probably represents transient TC interaction with the tRNA in a A/T state during a decoding step, the latter likely represents full and longer-lived accommodation of the tRNA into the A/A state.

The results outlined above validate the use of this assay to explore the effect of loop 3 mutations on ac-tRNA association with the IRES–ribosome complex independent of eEF2 activity. Mutants △1, △2, and △3 showed a progressive decrease in anisotropy (*Figure 5C*), following the trend established by the translation initiation and pseudotranslocation data. These data indicate that these mutants have a defect in initial ac-tRNA binding; in the case of △3, this defect is more severe than the effect of a non-cognate codon. This may be because the movement of the first codon into the A site has been compromised. ac-tRNA delivery to IRES–80S ribosome complexes with the △1 and △2 mutants was less than to WT, but equal to or greater than to the IRES with a non-cognate codon. To approximate the percentage of these A-site ac-tRNAs that successfully translocated to the P site, we normalized their P site binding levels to the A site interaction levels (*Figure 5D*). For △1, the percentage is ~80% while for △2 it is ~25%. When we consider these data in light of the proposed mechanism of IGR IRES-driven initiation (*Figure 1A*), they suggest that these mutants have defects in both pseudotranslocation events and these defects become progressively worse as loop 3 is shortened. In contrast, the G-rich and GGC mutants display ac-tRNA binding similar to the WT IRES (*Figure 5C*). Thus, the defect in these sequence mutants is restricted to the second pseudotranslocation event which moves ac-tRNA from the A site to the P site, and domain III from the P site to the E site. Taken together, the data from all mutants suggest that loop 3 has two independent functions

to facilitate two elongation factor-driven steps, which depend on loop 3 length and base composition.

## Loop 3 facilitates eEF2's ability to translocate ac-tRNA on IGR IRES–80S ribosome complexes

The anisotropy data show that loop 3 is important for initial ac-tRNA association with the ribosome, but do not directly address eEF2's role in this process. The decreased ac-tRNA association in mutant IRES–80S ribosome complexes observed in the anisotropy experiment could result from a decrease in spontaneous vacating of the A site, or from decreased TC association even if the A site is available. To address this, we used single-molecule total internal reflection fluorescence microscopy to directly visualize the colocalization of Cy5 fluorophore-labeled Phe-tRNA$^{Phe}$ with Cy3 fluorophore-labeled IRES–80S ribosome complexes (from yeast) that had been tethered (via the IRES RNA) to the surface of a microfluidic observation flowcell (*Figure 6—figure supplement 1*). This colocalization data reports on the ac-tRNA occupancy of the 80S–IRES ribosome complexes. We chose WT and △3 IRESs to study as they exhibited the most differing behaviors in the previous experiments. As expected, addition of just Phe-tRNA$^{Phe}$(Cy5)+GTP (without eEFs) to 80S–IRES ribosome complexes, followed by incubation and subsequent flushing of the flowcell to remove unbound ac-tRNA, revealed very low ac-tRNA occupancies for both WT and △3 IRESs (*Figure 6*). When GTP+eEF2 was included with the Phe-tRNA$^{Phe}$(Cy5) (but no eEF1A) the ac-tRNA occupancy of the IRES–80S ribosome complexes formed with WT IRES increased to $9.7 \pm 2.5\%$, consistent with a low, but enhanced level of eEF1A-independent ac-tRNA binding. When this experiment was repeated with the △3 IRES, we observed a lower ac-tRNA occupancy ($1.5 \pm 1.1\%$) compared to the WT IRES. Higher eEF1A-independent, but eEF2-dependent, ac-tRNA occupancy on WT IRES complexes compared to △3 IRES complexes suggests that the difference between these two IRESs in the anisotropy experiment (*Figure 5C*) is not due to altering eIF1A function. Rather, those data may indicate a decrease in clearing of the A site by the △3 mutant, suggesting the △3 mutant's main defect is in the first pseudotranslocation and not in the A-site ac-tRNA binding event itself.

To examine eEF1A-dependent ac-tRNA delivery, we assembled TC with Phe-tRNA$^{Phe}$(Cy5)+eEF1A+GTP and delivered this to the immobilized IRES–80S complexes without eEF2. Compared to the reactions lacking eEF1A, both IRESs show increased and similar ac-tRNA occupancies (WT: 17.9

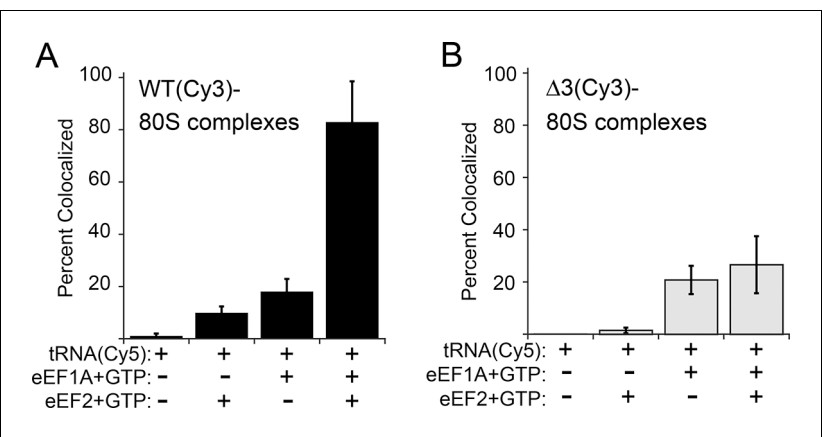

**Figure 6.** Effect of eukaryotic elongation factor 2 (eEF2) on colocalization of Phe-tRNA$^{Phe}$(Cy5) with individual 80S ribosome–internal ribosome entry site (IRES) complexes formed with either wild type (WT) (Cy3) IRES or △3(Cy3) IRES. Addition of elongation factors and Phe-tRNA$^{Phe}$(Cy5) (tRNA(Cy5)) to 80S ribosome–IRES complexes formed with either (**A**) WT(Cy3) IRES (black bars) or (**B**) △3(Cy3) IRES (gray bars) are depicted as percent Cy3-Cy5 colocalized spots. The presence or absence of factor(s) is indicated beneath the graphs and error bars represent one standard deviation from the mean. Elongation factors and ribosomes are from yeast.

The following figure supplements are available for Figure 6:

**Figure supplement 1.** Schematic of the single-molecule colocalization experiments.

± 4.8%, △3: 20.8 ± 5.4%). These data initially seem at odds with the anisotropy data in which eEF2-independent ac-tRNA association with 80S–WT IRES ribosome complexes is much greater than complexes with △3. This apparent discrepancy is likely due to the fact that anisotropy data are obtained under equilibrium conditions where transient interactions are observed, whereas the single-molecule fluorescence data are collected after the flowcell is flushed and thus only show stable long-lived association. Combining the data from both experiments reveals that eEF2-independent ac-tRNA association to WT IRES–80S ribosomes is transient and is inhibited by the △3 mutation.

Finally, when eEF2+GTP+TC was delivered to the tethered 80S–IRES ribosome complexes, we observed a dramatic increase in the ac-tRNA occupancy on complexes formed with the WT IRES (82.8 ± 15.7%), but not with the △3 IRES (26.6 ± 10.9%). This demonstrates that the △3 mutation inhibits the IRES–ribosome complex from using eEF2 to facilitate stable ac-tRNA delivery. Overall, our data suggest that loop 3 is important for eEF2's ability to catalyze both pseudotranslocations, the first of which moves domain III to clear the A site for ac-tRNA binding and the second which moves the first ac-tRNA to the P site.

### Comparison of results in lysate and reconstituted systems

Our toeprinting experiments performed in RRL and experiments conducted with reconstituted systems show some differences. Specifically, toeprinting with the G-rich and GGC mutants in RRL+CHX shows at least two rounds of translocation (*Figure 3A*) and at least four in RRL+ hygromycin B at low concentrations and post-treatment (*Figure 4B*). However, in the reconstituted assays these mutants fail before two rounds of pseudotranslocation (*Figure 5B*). We consider it unlikely that this discrepancy is due to differences in the species of ribosomes used (purified subunits were made from yeast and shrimp sources, versus rabbit subunits in RRL) because IGR IRESs function in diverse systems and contact highly conserved ribosome features. A more likely possibility is that the presence or effective concentrations of various components (ribosomes, ac-tRNAs, GTP, or unidentified factors) is different in the lysate as compared to the reconstituted system, which may alter the kinetics of the translocation reactions. In addition, the presence of antibiotics such as CHX or hygromycin B (which we only used in RRL-based experiments) may suppress the effects of sequence mutation to loop 3 by altering ribosome conformational dynamics (*Wilson, 2014*). Despite this uncertainty, taken together our data clearly identify loop 3 as important in more than one round of pseudotranslocation and also illustrate the importance of employing multiple experimental approaches.

## Discussion

To function, IGR IRESs must have affinity for the ribosome, promote subunit joining, manipulate elongation factor action, and move through the tRNA binding sites. In this study we show that conformationally dynamic loop 3 in the tRNA-mimicking domain controls two independent, non-canonical translocation events, demonstrating how a viral RNA can carry out intricate ribosome manipulation using dynamic RNA structure. This strengthens the previously postulated idea that structured regions are important for overall IRES architecture and ribosome positioning, whereas conformationally dynamic regions help drive the IRES through the ribosome in elongation factor-dependent steps to initiate translation (*Pfingsten et al., 2010*). The strategy of using a combination of conformationally flexible elements with stably structured domains is likely a strategy used by many RNAs that control dynamic cellular machines.

Our data show that the length and sequence of loop 3 are both important for function. A previous study also examined the effect of loop 3 length and sequence on IGR IRES translation efficiency (*Au et al., 2012*). The mutants in that complementary study showed modest defects in translation activity. However, toeprinting results showed that the position of domain III within the ribosome is similar, although differences in toeprint band intensity were sometimes observed. Overall, toeprint band intensity did not correlate well with translation activity, suggesting that something else regulates the modest defects that were identified in that study. Because we discovered mutants with more pronounced translation defects, and whose toeprint intensities did not correlate with translation activity, we could use this to more deeply dissect the specific mechanistic role of loop 3 in more depth using a battery of quantitative analyses. Our data indicate that domain III's loop 3 is involved in the two non-canonical pseudotranslocation events following initial IGR IRES recruitment of the 80S ribosome.

Although domain III was originally proposed to first bind in the P site, the most recent structural and mechanistic models, based on both additional structural information and reexamination of earlier published biochemical data, places domain III in the A site (*Figure 1A*) (*Fernández et al., 2014*; *Koh et al., 2014*; *Zhu et al., 2011*; *Muhs et al., 2015*). In this mechanistic model, initial pseudotranslocation by eEF2 is needed to clear the A site before ac-tRNA can bind the ribosome. Consistent with this, our data and other studies show that stable association of ac-tRNA with the IRES–ribosome complex depends on eEF2 (*Yamamoto et al., 2007*). Additionally, eukaryotic release factor 1 (eRF1) only binds in the A site of IRES–80S ribosome complexes (and induces a change in the toeprint) in the presence of eEF2 (*Jan et al., 2003*; *Muhs et al., 2015*). However, no pseudotranslocation is observed with pure WT IGR IRES–80S ribosome complexes treated with eEF2 only (assayed by toeprinting) (*Pestova, 2003*). A mechanistic model that reconciles this observation posits that eEF2 first moves domain III from the A site to the P site, but this is a transient state and without immediate ac-tRNA delivery domain III spontaneously reverse-translocates to the A site (*Fernández et al., 2014*). This is validated by the toeprinting experiment demonstrating one round of translocation in high concentrations of hygromycin B (*Figure 4A*), which has been shown to potently inhibit reverse translocation (*Borovinskaya et al., 2008*; *Szaflarski et al., 2008*). If this explanation is true, the transient position of domain III in the P site would preclude detection of this state by traditional biochemical approaches; possibly, the toeprinting assay itself may facilitate reverse-translocation. This mechanistic model is supported by our data and agrees with all previously published data.

Assuming domain III begins in the A site, shortening loop 3 appears to inhibit movement of domain III to the P site before any ac-tRNA is bound. Given that domain III and loop 3 are positioned to interact with components of the 40S subunit head known to be involved in translocation (ribosomal protein uS13 when domain III is in the A site, for example [*Cukras et al., 2003*]), our data favor a mechanistic model where the loop 3 length mutants fail to efficiently execute the first pseudotranslocation event and this blocks access of ac-tRNA to the A site. This is supported by the anisotropy data with the non-cognate RNA which show an increase above background levels established by the no-IRES control. This likely indicates the transient binding of the ac-tRNA TC to the A site and subsequent rejection. In comparison, the fact that the △3 mutant yields even lower anisotropy levels than the non-cognate RNA suggests that the TC can never bind the △3 IRES–ribosome complex even transiently. This is consistent with the idea that the initial movement of domain III does not occur with this mutant, either spontaneously or with eEF2, and domain III remains in the A site. Given that our sequence mutants (G-rich and GGC) inhibit the second pseudotranslocation, this interpretation makes loop 3, despite being a short and apparently conformationally dynamic element, a key player in non-canonical translocation events that move the IGR IRES through all three tRNA binding sites.

There is no obvious analogous structure to loop 3 in tRNA, raising the question of how this loop exerts its effects. One possibility is that loop 3 interacts directly with the ribosome in ways not yet clearly observed using structural methods. Recent cryoEM reconstructions of CrPV (*Fernández et al., 2014*) and Taura Syndrome Virus (TSV) (*Koh et al., 2014*) IGR IRESs bound to 80S ribosomes in the pretranslocated (PRE) state (domain III in the A site) at resolutions of 3.8 and 6 Å respectively and of CrPV–80S–eRF complexes in the post-translocation (POST) state (domain III in the P site) at 8.7 Å (*Muhs et al., 2015*) provide structural models for loop 3. However, the local resolution for loop 3 is low in all structures, consistent with conformational dynamics (*Figure 1C–E*). Interestingly, in the class I (CrPV) versus class II (TSV) IRESs, loop 3 spans somewhat different space when domain III is in the A site. In both structures, the 3' ends of loop 3 terminate in the decoding center of the A site where they may interact with elements of the decoding groove. In contrast, the 5' ends of loop 3 differ in these structural models. In CrPV the 5' nucleotides of loop 3 wrap around the 5' terminal nucleotides of the PKI stem in the A site. In the TSV structural model, loop 3 interacts with the apical loop of rRNA helix 24, part of a constriction between the P and E sites. In bacterial ribosomes this constriction is essential for maintaining the P-site tRNA in its proper place to prevent slipping of the mRNA (*Schuwirth, 2005*), and must be remodeled by 30S subunit head swiveling for tRNA to translocate from the P to the E site (*Zhou et al., 2013*; *Ratje et al., 2010*). If loop 3 contacts this constriction, it could affect a known structural regulator of translocation, affecting the conformation of the ribosome in a way that favors eEF2 function. In the POST structure with eRFs, loop 3 is modeled to interact with uS7, a key frame-maintenance and translocation regulator

(*Devaraj et al., 2009*; *Galkin et al., 2007*; *Robert and Brakier-Gingras, 2003*). Interestingly, the HCV IRES is also thought to communicate with uS7 (*Fukushi et al., 2001*; *Filbin et al., 2013*; *Boehringer et al., 2005*), pointing to this ribosomal protein as an important 'gatekeeper' to ribosome function that is exploited by viral IRES RNAs. Precisely what loop 3 interacts with, how and when it makes these interactions, and how these interactions affect the conformation of the IRES–ribosome complex remains to be determined, as does the question of whether loop 3 functions differently in the two classes of IGR IRESs.

In addition to making contacts to the ribosome, loop 3 could also affect pseudotranslocation by altering the conformational landscape of domain III, which comprises an H-type pseudoknot. Many H-type pseudoknots use adenosines in loop 3 to make minor groove interactions with an adjacent helix. Although no minor groove interactions have been identified in domain III, most IGR IRES loop 3s have adenosine content greater than 40% (*Figure 2—figure supplement 1*); this may be an important feature of loop 3. Indeed, the G-rich and GGC mutations (22% and 33% adenosine, respectively) show substantially decreased translation activity. Transient or dynamic interactions between the loop and the rest of domain III may be important for altering the conformation of the pseudoknot as it moves through the ribosome. tRNAs are known to undergo substantial conformational changes as they transit through the ribosome (*Dunkle et al., 2011*; *Fei et al., 2011*); loop 3 could help domain III do the same. Alternatively, it may be important for loop 3 to remain unstructured. Indeed, structural probing of these mutants in the unbound form show decreases in loop 3 accessibility to single-stranded ribonuclease (*Figure 4—figure supplement 1*). The presence and importance of these changes within the ribosome are unknown, although it is tempting to speculate that a decrease in flexibility may drive the defects observed in this study.

There is growing evidence that molecular mimicry is a common tool viruses use to infect their host cells; indeed, several plant viruses display tRNA mimicry in their 3' untranslated regions (UTRs) to enhance viral protein translation (*Dreher, 2010*; *Simon and Miller, 2013*). Yet, molecular mimicry is not limited to structural similarity; the binding partners of these mimics must also be fooled by conformational dynamics and overall molecular interactions. Our work suggests that the flexible elements of the IGR IRES facilitate these additional aspects of mimicry that remain understudied. This discovery that IRES RNA flexibility rather than defined structure is important for function may be particularly important in the context of ribosome manipulation since the ribosome has been suggested to act as a Brownian machine that fluctuates between conformational states (*Frank and Gonzalez, 2010*), and thus this and other elements of the translation machinery are highly tuned to respond to and exploit the dynamics of their ligands.

## Materials and methods

### Plasmid construction and cloning

The pCrPV1-1 dual LUC vector was a gift from Dr Eric Jan. Reporter vectors containing WT IAPV, *Homalodisca coagulata* Virus (HoCV), Kashmir Bee Virus (KBV), Himetobi P Virus (HiPV), TSV, *Solenopsis invicta* Virus-1 (SInV), and Acute Bee Paralysis Virus (ABPV) IGR IRES sequences were generated by polymerase chain reaction (PCR) amplification of the IRES sequence (plasmids were gifts from Dr Eric Jan and Dr Sunnie Thompson) and subsequent ligation into a dual LUC vector (pDBS, derived from pBluescript, a gift from Dr Les Krushel). Mutagenesis was employed using the Quik-Change (Agilent) method. DNA sequences encoding the RNA for assembly assays ('CrPV4': full IRES RNA sequence including GCU start codon) and RNase T1 probing ('CrPV11': domain III only, no start codon) were cloned into pUC19-derived vectors with a T7 promoter and a 5' Hammerhead ribozyme and 3' hepatitis delta virus (HDV) ribozyme flanking the IRES sequence. Constructs for reconstituted functional analysis ('FVKM RNAs') were built by PCR from the CrPV1-1 vector using primers that contained the appropriate mutations and flanked with restriction sites for cloning into pUC19 (without ribozymes). All cloned sequences including the LUC open reading frames were verified by standard sequencing methods using appropriate primers.

### RNA preparation

RNAs for translation assays were in vitro transcribed from XbaI-linearized vectors using the MEGA-script Kit (Life Technologies, Carlsbad, CA). RNA purification was performed by extraction with

TriReagent (Sigma, St. Louis, MO) followed by chloroform extraction and column purification using the RNeasy Kit (Qiagen, Germantown, MD) (*Plank et al., 2013*). RNAs for all other assays were made by in vitro transcription using T7 RNA polymerase and PCR-generated DNA templates, as described previously (*Pfingsten et al., 2007*). These RNAs were purified on 10% polyacrylamide-urea denaturing slab gels, passively eluted at 4°C, then concentrated and buffer-exchanged using appropriate MWCO centrifugal ultrafiltration devices (Millipore, Billerica, MA). All RNAs were assessed for quality using denaturing PAGE.

## Radiolabeling RNA and primers

RNAs not made with ribozymes were treated with rAPid alkaline phosphatase (Roche, San Francisco, CA) to remove the 5′ triphosphate, whereas no treatment was needed for RNAs made with a 5′ ribozyme or for synthetic primers (IDT, Integrated DNA Technologies, Coralvile, IA), which have a 5′ hydroxyl. RNA was 5′ end-labeled using T4 polynucleotide kinase (New England Biolabs, Ipswitch, MA) and $^{32}$P-gamma-ATP (PerkinElmer, Waltham, MA), then purified by denaturing gel electrophoresis, eluted, and precipitated as described previously (*Kieft et al., 1999*).

## In vitro translation assays

Pure dual LUC reporter RNAs were incubated in RRL (Promega, Madison, WI) supplemented with 150 mM potassium acetate (final concentration) and amino acids for 90 min at 30°C. LUC production was measured using the Dual Luciferase Reporter Assay System (Promega) and the GloMax Multi Detection plate reader. Data shown are from five independent experiments.

## mRNA degradation assays

Dual LUC reporter RNAs were body-labeled by including 1 µL of 50 µM (40 µCi total) $^{32}$P-alpha-UTP during transcription (described above), treated with TURBO DNase, and then desalted through G50 spin columns (GE Healthcare, Piscataway, NJ). Purified RNAs were diluted in nuclease-free water to 34,000 cpm/µL. Equal concentrations were verified by gel electrophoresis and phosphorimaging. For each time point, 2 µL of 34,000 cpm/µL dual LUC RNA were added to 8 µL of RRL and incubated at 30°C. These 10 µL reactions were collected at 0, 10, 30, 60, and 90 min, and were minimally processed by adding 30 µL of nuclease-free water and 40 µL of 2X urea loading buffer. Samples were kept on ice until 50 µL were electrophoresed on an 8% denaturing polyacrylamide gel (1 mm gel thickness) at 40 W for 1 hr and 45 min. The gel was wrapped in plastic and then exposed to a phosphorscreen at -20°C overnight. Phosphorscreens were imaged using a Typhoon scanner and data were analyzed in ImageQuant software by drawing equal sized boxes around the full length RNA at each time point and then normalizing data to the amount of signal in the time=0 sample for each RNA. Data were analyzed by linear regression analysis in Microsoft Excel.

## Toeprinting assay

For unbound IRES RNAs, 0.5 µg of toeprint RNA was mixed with 1.5 µL of 10X Toeprint Buffer A (1X: 20 mM Tris pH 7.5, 100 mM KOAc, 2.5 mM MgOAc$_2$, 2 mM dithiothreitol [DTT], 1 mM ATP, 0.25 mM spermidine), 0.5 µL of RNasin Plus (40 U/µL, Promega), and nuclease-free water to a final volume of 15 µL. For ribosome-bound RNAs (purified yeast 40S and 60S subunits or purified rabbit 40S), reactions were set up in the same way as above but included 8 pmol of each purified subunit. For RRL-incubated RNAs, 11 µL of RRL was pre-incubated with 1 µl of 45 mg/mL CHX or 1 µL nuclease-free water for 5 min at 37°C, and added to RNA and 10X buffer A as above. All reactions were incubated at 30°C for 5 min to allow for folding and binding. Then, 1 µL of 40,000 cpm/µL toeprint primer (internal photinus) and 24 µL of 1X Buffer A were added and incubated at 30°C for 5 min for primer annealing. Reverse transcription was performed by addition of 4 µL dNTPs (1.25 mM each), 1 µL 320 mM MgOAc$_2$, and 0.5 µL avian myoblastosis virus reverse transcriptase (25 U/µL, Promega) to each reaction. Primer extension proceeded at 30°C for 45 min, and was quenched with 4 µL of 4M NaOH and heated at 85°C for 5 min to hydrolyze RNA. Following this, 100 µl of nuclease-free water was added to each reaction before extraction with phenol:chloroform:isoamyl alcohol (PCIAA, 24:24:1, ThermoFisher, Waltham, MA), followed by CIAA (24:1) (ThermoFisher) extraction, and ethanol precipitation with 3 volumes of 100% ethanol and 1/10 volume of 3M NaOAc pH 5.3. Pellets were washed with 70% cold ethanol. Precipitated RNA pellets were dried and resuspended to equal

counts/μL in 1X TBE + 9M urea loading buffer, and then equal volumes (typically 10 μL) were loaded on a 10% polyacrylamide sequencing gel (0.4 mm gel thickness) with a sequencing ladder of the WT RNA (made by dideoxy-NTP incorporation as previously described; *Filbin et al., 2013*) and electrophoresed at 65 W for approximately 2 hr. Gels were dried and exposed to a phosphorscreen overnight; they were imaged on a Storm scanner (GE Healthcare) and analyzed in ImageQuant. 'Percent translocated' toeprints were calculated for each RNA in RRL with CHX treatment by quantifying the intensity of the +14/15 toeprint and the +20/21 toeprint in equal sized boxes in ImageQuant, and using these values in the equation: (+20/21)/(+14/15 + +20/21). Toeprinting assays using concentrated hygromycin B were performed essentially as described above; however 1 μL of 30 mg/mL hygromycin B (Roche) was added to the RRL and pre-incubated for 5 min at 37°C. For toeprinting assays in the presence of dilute hygromycin B, 0.5 μg of each RNA was incubated for 1 min in RRL/Buffer A/RNasin mix (as above) at 30°C before adding 1 μL of 0.05 mg/mL hygromycin B ('+') or nuclease free water ('-'). Reactions were incubated at 30°C for 5 min before adding radiolabeled primer and buffer as above. Reverse transcription and gel analysis were performed as described above.

## Ribosome and elongation factor purification

Both yeast (*Saccharomyces cerevisiae*) and shrimp (*Artemia salina*) eggs were used as sources of 40S and 60S ribosomal subunits. Yeast subunits were purified from strain YAS2488 (gift from J. Lorsch) as described (*Acker et al., 2007*). Briefly, cells were lysed using a liquid nitrogen mill, and clarified lysates were spun through 250 mM sucrose cushions under high-salt conditions to obtain clean 80S ribosomes. Subunits were separated by treatment with puromycin and resolved on 5–20% sucrose gradients. Crude shrimp egg 80S ribosomes were prepared from dried, frozen cysts as previously described (*Iwasaki and Kaziro, 1979*; *Thiele et al., 1985*) with some modifications. After the shrimp cysts were ground open, debris was removed by centrifugation at 30,000x*g* for 15 min and crude 80S ribosomes were precipitated from the supernatant by addition of 4.5% (w/v) polyethylene glycol (PEG) 20K according to previous methods (*Ben-Shem et al., 2011*). Subunits were resolved on 10–30% sucrose gradients after puromycin treatment. eEF1A was purified from yeast according to published methods (*Thiele et al., 1985*). His$_6$-eEF2 was isolated from an overexpressing yeast strain (TKY675; obtained from Dr Terri Kinzy), and purified as described (*Jørgensen et al., 2002*). Rabbit subunits were purified as described (*Kieft et al., 2001*).

## Tetrapeptide kinetics assay

Preinitiation complexes (Pre-ICs) were formed by incubation of shrimp egg 40S and 60S subunits with FVKM IRES RNA constructs at 37°C for 5 min in buffer 4 (40 mM Tris-HCl pH 7.5, 80 mM NH$_4$Cl, 5 mM MgOAc$_2$, 100 mM KOAc, 3 mM β-mercaptoethanol). tRNAs were charged with appropriate amino acids as described (*Pan et al., 2009*). Phenylalanine, valine, lysine, and $^{35}$S-methionine TCs with purified yeast eEF1A were formed as separate complexes by incubating the relevant charged tRNA (1.6 μM, based on amino acid stoichiometry) with eEF1A (8 μM) in buffer 4 supplemented with 1 mM GTP and 1 mM ATP at 37°C for 5 min. Tripeptide complexes were made by mixing Pre-ICs with 1 μM eEF2 and F, V, and K TCs at 37°C for 15 min. Using a quench-flow instrument, tetrapeptide complexes were made by mixing the tripeptide complexes with $^{35}$S-Met TC for defined time points on the millisecond scale. Reactions were quenched with 0.8 M KOH and peptide was released from tRNA by further incubation at 37°C for 3 hr. Samples were neutralized with acetic acid, lyophilized and suspended in water. Following centrifugation to remove particulates (which contained no $^{35}$S), the supernatant was analyzed by thin layer electrophoresis as previously described (*Youngman et al., 2004*). The identities of the tri- and tetrapeptides were confirmed by their comigrations with authentic samples obtained from GenScript (Piscataway, NJ). A further demonstration of tetrapeptide identity was provided by matrix-assisted laser desorption/ionization (MALDI) mass spectrometric analysis (Ultraflex III TOF/TOF, Bruker, Ewing, NJ).

## A-site tRNA binding: anisotropy

Phe-tRNA$^{Phe}$(prf) was prepared as previously described (*Wintermeyer and Zachau, 1974*; *Betteridge et al., 2007*). TC (0.1 μM, 250 μL) was incubated with shrimp 80S or shrimp 80S–IRES complex (0.1 μM, 250 μL) in buffer 4 for 15 min at 37°C and then kept on ice until anisotropy

measurement, which was performed at 23°C. Steady-state fluorescence anisotropy was determined using a Photon Technology International (PTI, Birmingham, NJ) QuantaMaster fluorometer with polarizer in L-format, with excitation at 462 ± 2 nm and fluorescence emission collected at 490 ± 2 nm. Instrument-integrated monochromators were used as filters for the fluorescence emission and the excitation light. The g-factor and anisotropy value were calculated using the instrument software as described (*Lakowicz, 1999*; *Ameloot et al., 2013*). The instrument was calibrated by using suspended nonfat dry milk aqueous solution as scatter. Experimental data were processed and analyzed by Felix software (from PTI).

## P-site tRNA binding: sucrose cushion cosedimentation

Shrimp 80S–IRES complexes containing Phe-tRNA[Phe] in the P site were formed by incubation of pre-IC (16 pmol) and Phe-TC (32 pmol) at 37°C for 15 min in the presence of 1 µM eEF2, in a total volume of 40 µL. The 80S–IRES complexes were isolated by ultracentrifugation at 4°C (540,000x$g$) for 40 min through a 1.1 M sucrose cushion, with 600 pmol of pure 30S bacterial ribosome subunits added as carrier to enhance pelleting and allow facile calculation of complex recovery. The pellets were gently washed twice with buffer 4 and dissolved in 100 µL of buffer 4 for $A_{260nm}$ determination. Recoveries typically varied between 60% and 80%. [3]H counts from the pellet were measured to determine the amount of [[3]H]-Phe-tRNA[Phe] bound to the complex.

## Translocation efficiency analysis

The percent A-site (*Figure 5C*) and P-site (*Figure 5B*) tRNA binding levels were each divided by the percent of A site binding for the WT, △1, and △2 mutants, and then multiplied by 100%. This permits analysis of the percentage of A-site tRNA that was moved to the P site for each of these RNAs.

## Single molecule colocalization assays

WT and △3 IRES RNAs for single-molecule analysis were generated with a 5′ extension of sequence (5′)-CA AAU CAA CCU AAA ACU UAC ACA-(3′) such that a complementary, 3′-biotinylated DNA oligo ((5′)-TGT GTA AGT TTT AGG TTG ATT TG/3Biotin/-(3′)) could be hybridized to the IRES constructs. The biotin at the 3′ end of the DNA oligo that had been hybridized to the IRES RNAs could then be used to tether the 80S–IRES ribosome complexes to the polyethylene glycol-, biotin-polyethylene glycol-, and streptavidin-derivatized quartz surface of a microfluidic observation flowcell (*Fei et al., 2008*; *Blanchard et al., 2004*; *Ha et al., 2002*). The 3′ end of the IRES RNAs contained one codon for Phe (UUC), followed by the hepatitis delta ribozyme to generate a clean 3′ end. 2′-3′ cyclic phosphates were removed as previously described (*Kieft et al., 1999*). IRES RNAs were labeled using Cy3-maleimide (GE Healthcare) and the 3′ DNA End-Tag Kit (Vector Labs, Burlingame, CA), which added one additional dG residue harboring the Cy3 label to the 3′ end of the IRES construct. IRES(Cy3) RNAs were purified from free dye by multiple phenol extractions and ethanol precipitation, or centrifugal filtration with a 10,000 Da MWCO (Millipore). Labeling efficiencies determined by $A_{260nm}$ and $A_{550nm}$ readings were typically low, ranging from 3% to 20%. A diagram of the RNA constructs is shown in *Figure 6—figure supplement 1*. Stocks of IRES(Cy3) RNAs that had been hybridized to the biotinylated DNA oligo were prepared by incubating a 10-fold excess (50 nM) of the 3′-biotinylated DNA oligo with either 5 nM WT IRES(Cy3) or 5 nM △3 IRES(Cy3) RNA (in a reaction volume of 100 µL) at 95°C for 2 min, slowly cooling the hybridization reactions to room temperature, transferring the hybridization reactions to ice, aliquoting, flash-freezing in liquid nitrogen, and storing the stocks at -80°C. These stocks, therefore, had 5 nM of either WT IRES(Cy3) or △3 IRES(Cy3) RNA.

Purified *Escherichia coli* tRNA[Phe] (Sigma) was fluorescently labeled with Cy5-NHS ester (GE Healthcare) at the primary aliphatic amino group of its naturally modified $acp^3U47$ residue, according to previously published protocols (*Fei et al., 2010*). The labeling reaction was quenched with 0.3 M NaOAc (pH 5.2), phenol-chloroform extracted, ethanol precipitated, and the Cy5-labeled tRNA[Phe] (tRNA(Cy5)) was separated from unlabeled tRNA[Phe] by hydrophobic interaction chromatography (HIC) using a TSK gel Phenyl-5PW column (Tosoh Biosciences, Tokyo, Japan) attached to an ÄKTA fast protein liquid chromatography (FPLC) system (GE Healthcare) as previously described (*Fei et al., 2010*). The HIC-purified tRNA[Phe](Cy5) was charged with phenylalanine (Sigma) as described using *E. coli* Phe-tRNA synthetase that was overexpressed and purified as previously

described (*Fei et al., 2010*). The charging reaction was quenched with 0.3M NaOAc (pH 5.2), phenol-chloroform extracted, ethanol precipitated, resuspended in 10 mM ice-cold KOAc (pH 5), passed through a Micro Bio-Spin Gel Filtration spin-column (Bio-Rad, Hercules, CA), aliquoted, flash-frozen in liquid nitrogen, and stored at -80°C. Charging efficiency was estimated by running an aliquot through a Phenyl-5PW column to detect the charged Cy5-Phe-tRNA^Phe and uncharged Cy5-tRNA^Phe, separated by HIC. The typical charging efficiency in these reactions was >90%.

For each colocalization experiment, IRES–80S ribosome complexes were initially assembled using 1.25 nM oligo-hybridized-Cy3-IRES RNA and 100 nM each of yeast 40S and 60S subunits in 1X Eukaryotic Polymix Buffer (EPB: 50 mM Tris-acetate at pH 7 at 25°C, 100 mM KOAc, 10 mM MgOAc$_2$, 0.5 mM spermidine, and 10 mM β-mercaptoethanol). In a separate reaction tube, a TC was prepared using 500 nM Phe-tRNA^Phe(Cy5), 5 μM eEF1A, and 2 mM GTP in 1X EPB. Each of these two reaction tubes were incubated at 37°C for 10 min. Then, 1 μM eEF2 and 2 mM GTP were added to the IRES–80S ribosome complex to initiate the first pseudotranslocation reaction and the reaction was allowed to proceed for an additional 10 min at 37°C (during which the reaction tube containing the TC was kept on ice). Subsequently, the TC was added to the IRES–80S ribosome complex (containing eEF2 and GTP) and the entire reaction incubated for another 10 min at 37°C. Finally, the entire reaction was diluted fivefold in 1X EPB and the diluted reaction was delivered into the polyethylene glycol-, biotin-polyethylene glycol-, and streptavidin-derivatized quartz microfluidic observation flowcell (*Blanchard et al., 2004*). The 80S–IRES ribosome complex was incubated in the flowcell for 5 min and components that remained untethered to the surface of the microfluidic flowcell at the conclusion of the 5 min were washed out of the flowcell using an imaging buffer composed of 1X EPB and a protocatechuic acid/protocatechuate-3,4-dioxygenase based oxygen scavenging system (*Aitken et al., 2008*). Cyclooctatetraene (COT, Sigma) and 0.012% v/v 3-Nitrobenzyl alcohol (NBA, Sigma) were included as triplet state quenchers in these experiments.

Surface-tethered, Phe-tRNA^Phe(Cy5)-bound 80S–IRES ribosome complexes were imaged using a custom-built, prism-based total internal reflection fluorescence microscope. Cy3 and Cy5 fluorophores were excited with a 532 nm laser and a 640 nm laser, respectively, with their powers attenuated such that the laser beams measured ~8 mW when they hit the prism. Emission data were directed to the image sensor of an electron-multiplying charge-coupled device (EMCCD) camera that records the fluorescence emission as a ~2 min movie with a frame rate of 100 msec. Prior to striking the image sensor of the EMCCD camera, the fluorescence emission from Cy3 and Cy5 are wavelength-separated using dichroic beamsplitters such that they could be directed onto the two separate halves of the image sensor. Colocalization data were analyzed from the imaged frames, using the standard software MetaMorph, as follows: the 256 pixel x 256 pixel imaged frames were split into the green and red halves, each half being 128 pixel x 256 pixel. Spots were picked from the red frame, using automated features in MetaMorph® and designated as 'Areas'. The red frames were then stacked on the green frames and the 'areas' were transferred from the red to the green frames. Automated algorithms set thresholds to the intensities, assigned geometric coordinates to the spots, calculated the spread of each spot intensity over an average of four adjacent pixels, superimposed each Cy5 frame on the corresponding Cy3 frame and calculated the number of spots that showed significant spatial overlap. This analysis is performed on every frame of the movie captured for a given reaction condition.

For the experiments designed to test the effect that the absence of eEF2, prior to addition of the TC, had on the colocalization, the first 10-min incubation step of the IRES–80S complex with eEF2-GTP was omitted. For these experiments, after imaging the IRES–80S complexes with Phe-tRNA^Phe(Cy5) delivered by eEF1A, the same channel was washed three times with 1X EPB to remove all unbound components, and a fresh mix of pre-incubated eEF2-eEF1A-GTP-Phe-tRNA^Phe(Cy5) was delivered to the flowcell prior to a second round of imaging aimed at monitoring the rescue of colocalization by addition of eEF2. Similarly, in experiments targeted to detect the effect of eEF1A on colocalization, eEF1A was not added to the initial reaction tube in which the TC was set up. In this case, after imaging the IRES–80S complexes with Cy5-Phe-tRNA^Phe, the channel was washed with 1X EPB, and a fresh mix of pre-incubated TC containing eEF2-eEF1A-GTP-Phe-tRNA^Phe(Cy5) was delivered to the flowcell to detect restoration of colocalization. All experiments were performed at least in duplicate and data from at least five movies for each experiment were averaged to calculate the colocalization percentage under a given set of conditions.

## Assembly assays

In 30 mM HEPES-KOH pH 7.5 and 10 mM MgCl$_2$, 1000 cpm of 5' end-labeled CrPV3 RNAs (IRES alone, no coding sequence) were folded by heat-cooling. Folded RNAs were incubated at 37°C in 30 μL RRL containing 1.2 mg/mL hygromycin B for 20 min. All samples were diluted in 500 μl ribosome association dilution buffer (RADB, 50 mM Tris pH 7.5, 50 mM NaCl, 5 mM MgCl$_2$, 1 mM DTT) and separated by 15–30% sucrose gradient density fractionation in an SW41 rotor for 3 hr at 36,000 rpm, 4°C. Fractions were collected on a BioComp gradient maker and fractionation system. The amount of $^{32}$P in each fraction was determined by filter binding and exposure to a phosphorscreen.

## Filter binding assays

### Approximate on-rate

IRES RNAs and a negative control RNA (Murray Valley Encephalitis Virus xrRNA) were 5' end-radiolabeled. The RNAs were diluted to 100 cpm/μl in RNase-free water, which resulted in RNA concentrations in the attomolar range; 100 cpm of RNA was used per 50 μL reaction. RNAs were heated at 85°C for 1 min in 30 mM HEPES-KOH pH 7.5 and removed from heat. To 10 mM final concentration, MgCl$_2$ was added and the RNAs were allowed to cool on the benchtop for 5 min. Pure shrimp ribosomes were added to the RNA at room temperature to a final concentration of 30 nM, and then 50 μL aliquots were removed from the reaction at defined time points out to 12 min and immediately pipetted through a membrane sandwich of nitrocellulose (BioRad) (on the top), Hybond nylon membrane (GE Healthcare) (middle), and Whatman filter paper (VWR, Radnor, PA) (bottom), on a dot-blot vacuum manifold. Membranes were air-dried then exposed to a phosphorscreen. The screens were imaged on a Typhoon phosphorimager scanner. The data were analyzed by drawing equal sized circles around each dot using ImageQuant software and obtaining a volume/intensity report for each circle. Fraction bound was then calculated from the intensity signals as follows: (Nitrocellulose)/(Nitrocellulose + Nylon).

### Approximate off-rate

100 cpm of RNA per 50 μL reaction was folded as described above in 30 mM HEPES-KOH pH 7.5 and 10 mM MgCl$_2$. Then, 15 nM purified yeast 40S and 60S ribosomal subunits were added to the folded RNA and incubated at 37°C for 15 min. Following this, 5 μg of unlabeled RNA was added to each reaction (WT RNA added to the WT reactions, and G-rich RNA added to the G-rich reactions, ~240 nM), and 50 μL aliquots were removed at defined time points out to 30 min and immediately applied to the membrane sandwich as described above. Data were analyzed as described above.

### RNase T1 probing

40,000 cpm of 5' end-radiolabeled CrPV11 (domain III only) WT, G-rich, and △3 RNAs were folded by heat-cooling in 30 mM HEPES-KOH pH 7.5, 10 mM MgCl$_2$, in the presence of 1 μg carrier tRNA. RNase T1 (Roche) digestion was performed by adding 0.1 U of enzyme and incubating at 37°C for 2 min. RNAs were ethanol precipitated overnight and resuspended to equal counts per microliter in 1X TBE + 9M urea loading buffer. RNase T1 (G) (denaturing) ladders for each RNA and a hydrolysis ladder of the WT CrPV11 RNA were generated as previously described (*Kieft et al., 1999*). Samples were loaded on a 12% polyacrylamide-urea sequencing gel (0.4 mm gel thickness) and run for 2 hr at 65 W. For analysis, data were normalized to total amount of radiation in each lane before subtracting the appropriate non-native T1 cleavage signal (G ladders) from the native T1 cleavage signal.

## Acknowledgements

We thank Eric Jan, Nobuhiko Nakashima, Sunnie Thompson, and Anne Willis for plasmids encoding IRESs and LUCs. Erik Hartwick assisted with eEF purification. We thank Thomas Evans, Eric Jan, Megan Filbin, and David Costantino for critical reading of this manuscript and members of the Kieft Lab for useful discussions. This work was supported by grant GM-097333 (to JSK) and GM-080376 (to BSC) and Burroughs Wellcome Fund (CABS 1004856), R01 GM-084288, and a Camille Dreyfus Teacher-Scholar Award (to RLG). JSK is an Early Career Scientist of the Howard Hughes Medical Institute. MDR was supported as an American Heart Association Predoctoral Fellow

(12PRE11900057) and through NIH T32 training grant GM-008730. SM was supported by a Susan G. Komen for the Cure ® Postdoctoral Fellowship (PDF12231199).

## Additional information

### Funding

| Funder | Grant reference number | Author |
| --- | --- | --- |
| National Institutes of Health | GM-097333 | Jeffrey S Kieft |
| American Heart Association | 12PRE11900057 | Marisa D Ruehle |
| Howard Hughes Medical Institute | Early Career Scientist Award | Jeffrey S Kieft |
| National Institutes of Health | GM-080376 | Barry S Cooperman |
| Burroughs Wellcome Fund | CABS 1004856 | Ruben L Gonzalez |
| National Institutes of Health | GM-084288 | Ruben L Gonzalez |
| Camille and Henry Dreyfus Foundation | Teacher-Scholar Award | Ruben L Gonzalez |
| National Institutes of Health | Training Grant GM-008730 | Marisa D Ruehle |
| Susan G. Komen for the Cure | PDF12231199 | Somdeb Mitra |

The funders had no role in study design, data collection and interpretation, or the decision to submit the work for publication.

### Author contributions

MDR, Conception and design, Acquisition of data, Analysis and interpretation of data, Drafting or revising the article; HZ, Conception and design, Acquisition of data, Analysis and interpretation of data; RMS, YC, Acquisition of data, Analysis and interpretation of data; SM, Acquisition of data, Analysis and interpretation of data, Drafting or revising the article; RLG, Analysis and interpretation of data, Drafting or revising the article; BSC, JSK, Conception and design, Analysis and interpretation of data, Drafting or revising the article

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
