## [Decision Letter]

Thank you for submitting your work entitled" A dynamic RNA loop in an IRES affects multiple steps of elongation factor-mediated translation initiation" for peer review at *eLife*. Your submission has been favorably evaluated by Michael Marletta (Senior editor), a Reviewing editor, and two reviewers.

The reviewers have discussed the reviews with one another and the Reviewing editor has drafted this decision to help you prepare a revised submission.

The manuscript of Ruehle et al. describes comprehensive biochemical investigation of a key RNA element in intergenic region (IGR) internal ribosome entry site (IRES) mediated translation initiation. These IRES elements, whose founding member is the cricket paralysis virus IRES (CrPV IRES), bind directly to ribosomes from a range of organisms, drive subunit assembly and can initiate elongation in the absence of an initiator tRNA. In particular, the authors focus on a key loop within a pseudoknot that mimics the codon-anticodon interaction in the IRES sitting initially in the ribosomal A site in the 80S complex; as such the IRES requires translocation (pseudotranslocation) to move the IRES to the P site to allow elongation. Here the authors use a wide variety of experimental approaches – toeprinting, dual luciferase assays, co-sedimentation assays, fluorescence anisotropy and single-molecule co-localization assays – to probe the role of the loop. They argue that certain mutations in the loop affect the initial pseudotranslocation step while others affect the 2nd pseudotranslocation step. In general, the conclusions of the work are supported by the array of experiments, and the mechanistic model is clear and substantiated. Moreover, these studies nicely shed light on how dynamic features of RNA elements are critical is specifying their unusual function on the ribosome.

Despite the overall strengths of the manuscript, there are a number of issues that the authors need to address in order to make the manuscript suitable for *eLife*. Broadly speaking, the presentation of the manuscript muddles the story in a number of places and creates some confusion.

First, the authors here are probing the function of loop 3 of the CrPV IRES, following on earlier work by Jan and colleagues, and they reach quite different conclusions. It is important for the authors to discuss these differences. The authors also make some broad statements about the consequences of loop 3 mutations on complex formation and that are difficult to reconcile with some of the toeprinting data. For example, the authors broadly conclude that there are no defects in complex formation that result from the various loop 3 mutations (based on Figure 4—figure supplement 1) and yet there are clear differences in the level of toeprint that is observed; differences in these results should be fairly discussed. There are a number of similar discrepancies that should be addressed throughout the manuscript (see detailed points).

Another major point of concern is the many different systems used throughout the manuscript. While the authors would like to claim that all the eukaryotic components are compatible with one another (from yeast to shrimp to rabbit etc), we know from an earlier publication where CrPV is characterized on *E. coli* ribosomes that mechanistic differences can result. At a minimum, the authors should clarify throughout the manuscript when heterologous components are utilized, and that there are some cautions that need to be considered. Better yet, the authors could run toeprinting experiments on each system to show that rudimentary features are conserved throughout the species utilized.

Finally, there are a number of places where additional controls would strengthen the conclusions. For the luciferase assays, it should be demonstrated that the mRNAs are intact and equivalently abundant. For the binding assays, the dependence of high background binding (20%) should be sorted out by providing additional controls (minus ribosomes, minus 60S, etc.).

*Reviewer #1*:

The manuscript by Ruehle et al. describes a mutation analysis of RNA loop 3 that connects the anticodon- and codon-like elements within the pseudoknot I (PKI) domain of the IGR IRES of the *Dicistroviridae* viruses. Their data show that both the length and the sequence of this loop determine the efficiency of translation from different types of IGR IRESs. Specifically, they found, using the IGR IRES from Cricket Paralysis Virus (CrPV), that the intactness of loop 3 is important for both elongation factor-dependent pseudotranslocation events. Since loop 3 is unlikely to directly interact with elongation factors, they put forward a hypothesis in which this loop assists in pseudotranslocation by affecting ribosome conformation. Interestingly, hygromycin B was found to inhibit the first but not the second translocation event driven by the CrPV IRES. Overall, if sufficiently developed, this study will provide important information about the features of IGR IRESs that contribute to their efficiency in translation at the steps downstream of the initial IRES-80S ribosome interaction. However, there are major issues with the toeprint assays that they carried out to support their main conclusions.

Major comments:

1) An earlier mutagenesis study by Au and Jan (2012) demonstrated that IGR IRESs tolerate significant changes in the loop-3 length, which at sharp variance with the results of the present study. Some explanation is required.

2) Figure 4. It seems that WT mutant IRESs recruit yeast 80S ribosomes with different efficiencies, which casts doubt on their conclusion that "Domain III is dispensable for efficient 40S and 60S binding". In addition, the relative intensities of these toeprints poorly correlate with the formation of ribosomal complexes resolved by centrifugation (Figure 4—figure supplement 1). Why? In fact, for Figure 4, presenting pretranslocation toeprints with RRL in which eEF2 is specifically inhibited by diphtheria toxin would be more appropriate than those with yeast ribosomes.

3) Figure 4 shows that the delta 3 mutation although inhibitory does completely block translocation. This is inconsistent with the complete absence of the 20/21 toeprint either in the presence or absence of cycloheximide (lane 16, panel A). There might be a mistake in measuring band intensities/calculations.

4) Why does the GGC mutant binds 80S in Figure 4 (lane 26), but not in Figure 4—figure supplement 1?

5) In Figure 4—figure supplement 1 there is a significant binding of IRESs to the 40S ribosome in RRL. Therefore, it is surprising that they don't see 40S toeprints in Figure 4. Are these coinciding with the 80S toeprints? A control with GMPPNP, which inhibits 60S binding, would be helpful in resolving this issue.

6) “…the length mutants do not execute the first pretranslocation event…”. Although the toeprinting experiments using hygromycin B seem to support this conclusion (Figure 5), those done with the delta-1 and delta-2 mutants using cycloheximide tell the opposite (Figure 4). Please reconcile these results.

*Reviewer #2*:

The manuscript of Ruehle et al. describes comprehensive biochemical investigation of a key RNA element in intergenic region (IGR) internal ribosome entry site (IRES) mediated translation initiation. These IRES elements, whose founding member is the cricket paralysis virus IRES (CrPV IRES) bind directly to ribosomes from a range of organisms, drive subunit assembly and can initiate elongation in the absence of an initiator tRNA. In particular, they focus on a key loop within a pseudoknot that mimics the codon-anticodon interaction in the IRES sitting initially in the ribosomal A-site in the 80S complex; as such the IRES requires translocation (pseudotranslocation) to move the IRES to the P site to allow elongation. Here the authors use a wide variety of experimental approaches–toeprinting, dual luciferase assays, co-sedimentation assays, fluorescence anisotropy and single-molecule co-localization assays to probe the role of the loop. They show that certain mutations in the loop affect the initial pseudotranslocation step while others affect the 2nd pseudotranslocation step. In general, the conclusions of the work are supported by the array of experiments, and the mechanistic model is clear and substantiated. However, the presentation of the manuscript, as written, muddles the story and creates confusion. This manuscript would merit publication in *eLife* after the authors address this global concern and points outlined below.

1) The work presented uses many different translation systems–yeast, shrimp, *E. coli,* rabbit retic and insect viruses. The need to utilize reagents from various origins is understandable and unavoidable due to inefficiencies of the current reconstituted systems, but a certain rigor is needed. I counted at least 4 mixed translation systems in this paper. Furthermore, tRNA can be from yeast or *E. coli*; similar true for the ribosomes that could be yeast or shrimp. It creates confusion, as sometimes it is impossible to deduce what combination of reagents has been used. It would be desirable if authors clearly indicate sources of ribosomes, factors and tRNA for every experiment.

While it is fully acceptable to use mixed model systems, a caution should be used when such systems are compared to each other. Curiously, Kieft et al. recently published a paper where it was demonstrated that translation initiation on CrPV IRES occurs via different mechanism in *E. coli*. The authors should show that initiation in the various mixed systems used in the current manuscript occurs via similar mechanisms. Perhaps a toeprint in each used system, as the paper already presents a toeprint for yeast ribosomes in RRL. It would be useful if authors would explain why they feel a need to use multiple systems, as it will help to understand how comparable those systems are. This is the biggest weakness of the current work, and one the authors should be able to readily address, given the robust mechanism of the IGR IRESs.

2) Dual luciferase assay data should be supported by experiments demonstrating reporter mRNA integrity and abundance. This is especially true when different reporter constructs are compared, and more so when changes are expected to affect mRNA structure. Northern blot of RRL samples used to measure luciferase activity would be highly preferable, if that is impossible authors should show the denaturing and native gel of the reporter mRNAs.

---

## [Author Response]

*[…] First, the authors here are probing the function of loop 3 of the CrPV IRES, following on earlier work by Jan and colleagues, and they reach quite different conclusions. It is important for the authors to discuss these differences.*

The Au and Jan paper is a very valuable contribution to the field and we have revised both the Results and Methods to better discuss the different conclusions we make and the basis for this difference. The source of the difference is an important point that bears on several reviewer comments, and thus in addition to our revision in the manuscript itself, we feel it is worth devoting some explanation here. Simply stated, our conclusions differ not because of conflicting data, but largely because of how we interpret what the toeprinting experiment can and cannot reveal.

To summarize the relevant data of Au and Jan (2012), the authors altered loop 3 length and sequence using translation assays and the toeprinting experiment. They found that increases in loop 3 length were tolerated functionally up to four additional nucleotides. We did not assess the effect of increasing the length of loop 3, which we did not think would be as mechanistically insightful. The effect of decreasing loop 3 length that the Jan group tested mirrored our results, although they deleted only one or two nucleotides from the loop, not three as we did. In toeprinting assays, these deletion mutations generated a pretranslocation toeprint at the same position as the WT RNA using the antibiotic edeine as an inhibitor in the lysate. Notably, the toeprint intensity varied from 61-126% of WT, thus toeprint intensity did not correlate with translation activity. This is consistent with our data. In addition, Au and Jan present a large number of point mutations in their paper analyzed the same way, with varying levels of activity and toeprint intensities. Au and Jan conclude that their mutations alter 80S binding efficiency and that “ribosome positioning” is affected by these mutations, although the position of the toeprint does not change in their experiments. Likewise, they do not include alternate approaches to directly measure ribosome binding.

Our interpretation of the toeprinting data is different, based on our assessment of the method’s strengths and weaknesses after applying it to several IRESs over many years. Our assessment is that while the method is powerful, the results are often over-interpreted or used as an inappropriate proxy for more quantitative methods. To illustrate, in Figure 4 of our manuscript two toeprint experiments are shown with WT IRES RNA, performed on the same day. There is variability in the intensity of the toeprints even though the position of the toeprints is the same. This is likely due to experimental variables such as different extension and cDNA extraction efficiencies as well as differences in how the RT might stop due to conformational dynamics of the ribosome, the dynamic nature of the system, etc. Thus, a quantitative analysis requires normalization and averaging of multiple replicates (as we do in Figure 4). However, even with this careful quantitation, we are convinced that the results should not be interpreted as a quantitative measure of ribosome binding without additional complementary approaches (as we do in Figure 4—figure supplement 1). Thus, we assert that toeprint intensity is not a quantitative measure of ribosome binding or complex formation, it is best used to monitor the position of bound ribosomes on the mRNA. This is part of the reason why we regard the toeprinting data as one part of a much more comprehensive examination using a “wide variety of experimental approaches–toeprinting, dual luciferase assays, co-sedimentation assays, fluorescence anisotropy and single-molecule co-localization assays” as pointed out by a reviewer. Furthermore, because we discovered mutants with more pronounced translation defects and applied this set of novel and more quantitative biochemical assays designed to interrogate each step in the initiation pathway, we could dissect the specific mechanistic role of loop 3 to a much greater depth.

In summary, while the raw data presented here are not at odds with the valuable data that Au and Jan reported, we must “agree to disagree” regarding interpretation, and thus the conclusions we reach are slightly different. With all this pointed out, we fully agree that the basis of our interpretation and the resultant differences needed to be better articulated. Thus, we now include text that explicitly states the way in which we use toeprinting data.

*The authors also make some broad statements about the consequences of loop 3 mutations on complex formation and that are difficult to reconcile with some of the toeprinting data. For example, the authors broadly conclude that there are no defects in complex formation that result from the various loop 3 mutations (based on Figure 4—figure supplement 1) and yet there are clear differences in the level of toeprint that is observed; differences in these results should be fairly discussed.*

We agree that the way we presented and discussed our data caused confusion and did not make the point well. We have therefore revised the manuscript and added additional data as follows:

1) The statement “domain III is dispensable for efficient 40S and 60S binding” comes from published data (Nishiyama et al., 2003; Costantino and Kieft, 2005; Jan and Sarnow, 2002). These studies showed that removal of the entire domain III has no effect on the ability of the remaining IRES RNA to assemble 80S complexes. Thus, we expect that smaller mutations to the domain would also not affect binding. We have clarified this statement in the text.

2) We now better explain the use of several complementary approaches to assess 80S complex formation. As explained above, we now more clearly point out that we do not use toeprinting as a quantitative measure of the efficiency of complex formation. We also call better attention to the direct and quantitative filter binding experiment that shows no difference in the approximate on- and off-rates of the WT and mutant IRES binding to ribosomes. In addition, we replaced Figure 4—figure supplement 1 (shows 80S formation on IRESs by sucrose gradient) with a much better version in which the RNA constructs used were more appropriate and the reaction was captured using an antibiotic. The conclusions that these results support are more clearly described in the text.

*There are a number of similar discrepancies that should be addressed throughout the manuscript (see detailed points). Another major point of concern is the many different systems used throughout the manuscript. While the authors would like to claim that all the eukaryotic components are compatible with one another (from yeast to shrimp to rabbit, etc.), we know from an earlier publication where CrPV is characterized on* E. coli *ribosomes that mechanistic differences can result. At a minimum, the authors should clarify throughout the manuscript when heterologous components are utilized, and that there are some cautions that need to be considered. Better yet, the authors could run toeprinting experiments on each system to show that rudimentary features are conserved throughout the species utilized.*

We appreciate the confusion that might arise from using heterologous systems and that we did not do a good job of defending or motivating their use. We have adjusted the manuscript in several ways:

1) We now point out the abundance of studies of the IGR IRESs using components from a variety of sources and from mixtures of these (yeast, rabbit, wheat germ, shrimp, pig, human, etc.), and most importantly that the conclusions from these systems are identical and consistent. A set of representative references are included to show that the literature contains strong, established, and widely accepted evidence that the IRES works identically in all of these eukaryotic and in heterologous systems. In addition, although the figure legends specified the species of ribosomes/elongation factors or lysate that was used, we also now include that information in the text as much as possible.

2) We now include additional supplementary data, which is a toeprinting analysis of the WT IRES bound to yeast subunits and pure rabbit subunits (Figure 4—figure supplement 2). The position of the toeprints is the same and matches published toeprints from a variety of papers using diverse sources. When combined with the wealth of published literature this justifies the use of diverse systems.

3) In regard to the question of IGR IRES activity in *E.coli*, we regret the understandable confusion that this caused. That study (from our lab) showed that the IRES can bind the bacterial ribosome largely by shape-complementarity to the decoding groove, that the mRNA moves by some mechanism to an AUG downstream to initiate translation using a process *not* dependent on the same structure-based mechanism that is known to drive efficient translation in all eukaryotes studied. Hence, the mechanisms are not comparable and that study does not change the way we understand the mechanism in eukaryotes. In our study, we used eukaryotic ribosomes and elongation factors from organisms that have previously been used to study IGR IRES function and which have shown mechanistic consistency. We now more clearly point this out in the manuscript.

*Finally, there are a number of places where additional controls would strengthen the conclusions. For the luciferase assays, it should be demonstrated that the mRNAs are intact and equivalently abundant.*

This is an astute and valid concern. Two points that address this are now included in the text:

1) Since we used an in vitro lysate system we precisely control the amount of input RNA, which we kept constant. The upstream RLUC reporter provides an internal control to help verify this.

2) We assessed whether the different mRNAs were indeed intact and whether their degradation rates were altered by mutation. Because our assays were done using cell lysate, we used the straightforward technique of body-radiolabeling input reporter mRNAs and then incubating them in rabbit reticulocyte lysate over the time course and temperature of the translation experiments. These data are now shown in Figure 2—figure supplement 1, which shows that the different mutants behave the same. Thus mRNA quality or abundance in the translation assays do not account for the differences in IRES activity that we observed.

*For the binding assays, the dependence of high background binding (20%) should be sorted out by providing additional controls (minus ribosomes, minus 60S, etc.).*

This is a good suggestion. To adjust the manuscript we included additional data and discussion:

1) The minus ribosome control was conducted but was not included in the original manuscript. These data are now provided in Figure 6—figure supplement 3. Briefly, the data clearly show increases in raw anisotropy signal as we move from free tRNA, to eEF1A- bound tRNA (ternary complex, TC), to TC+80S (no mRNA), and finally TC+80S+IRES. There is background association of tRNA to empty 80S ribosomes, which is expected.

2) The minus 60S control was not performed because this would change the extent of the mass shift that causes the altered anisotropy signal, and this would not be readily interpreted with the other data. This control would only assess binding of tRNA to 40S subunits which is neither stable nor directly relevant to our study.

3) We include the more stringent control of measuring tRNA binding to 80S complexes without an IRES RNA present and compare that to IRES-80S complexes with cognate or noncognate codons. These data show background tRNA binding to 80S complexes (no mRNA control), the specific influence of using an IRES with a cognate codon that should more stably associate with tRNA, and the transient binding caused by proofreading and rejection of tRNA (non-cognate control). We better point out the importance of this control in the manuscript text.

Reviewer #1:

Major comments:

*1) An earlier mutagenesis study by Au and Jan (2012) demonstrated that IGR IRESs tolerate significant changes in the loop-3 length, which at sharp variance with the results of the present study. Some explanation is required.*

Please refer to the response above.

*2) Figure 4. It seems that WT mutant IRESs recruit yeast 80S ribosomes with different efficiencies, which casts doubt on their conclusion that "Domain III is dispensable for efficient 40S and 60S binding". In addition, the relative intensities of these toeprints poorly correlate with the formation of ribosomal complexes resolved by centrifugation (Figure 4—figure supplement 1). Why? In fact, for Figure 4, presenting pretranslocation toeprints with RRL in which eEF2 is specifically inhibited by diphtheria toxin would be more appropriate than those with yeast ribosomes.*

The response above regarding our interpretation of toeprinting and the use of complementary approaches relates to this point, and we refer the reviewer to that.

In addition, in regard to the reviewer’s comment about inhibiting the initial translocation event to capture the pretranslocation toeprint using diphtheria toxin, we have performed toeprinting with pure rabbit and yeast ribosomes and observed the same toeprint with both (now in Figure 4—figure supplement 3). This, by definition, represents the pretranslocated position as no other factors are present and because it is in the same position regardless of the species of ribosome, it is a “marker” of the initial pretranslocated binding location. Furthermore, we also show the location of the pretranslocation toeprint by using hygromycin B in RRL (Figure 5). Our sequencing ladders are of high enough quality to allow precise location of the toeprint; these pretranslocation toeprints are always in the same location.

3) Figure 4 shows that the delta 3 mutation although inhibitory does completely block translocation. This is inconsistent with the complete absence of the 20/21 toeprint either in the presence or absence of cycloheximide (lane 16, panel A). There might be a mistake in measuring band intensities/calculations.

We thank the reviewer for this astute observation. This resulted from a lack of background signal subtraction for each RNA during the image analysis, which is why the delta 3 mutation appears to be higher than it should be and the WT appears to be lower than it should be. The data were re-analyzed to subtract this background signal and have now replaced the original figure.

4) Why does the GGC mutant binds 80S in Figure 4 (lane 26), but not in Figure 4—figure supplement 1?

We believe the reviewer was referring to the GGC mutant in the toeprinting experiment, but to the G-rich mutant in the assembly assay. The assembly assay originally shown in Figure 4—figure supplement 1 showed the GGC mutant to assemble 80S complexes at similar levels as WT, whereas the G-rich mutant had a lower peak. To address this, as stated above we have replaced Figure 4—figure supplement 1 with a much better version that more clearly shows robust 80S assembly. Also, the issue of toeprint band intensity as a representation of 80S binding efficiency is discussed above.

*5) In Figure 4—figure supplement 1 there is a significant binding of IRESs to the 40S ribosome in RRL. Therefore, it is surprising that they don't see 40S toeprints in Figure 4. Are these coinciding with the 80S toeprints? A control with GMPPNP, which inhibits 60S binding, would be helpful in resolving this issue.*

The location of 40S and 80S toeprints are the same and thus binding of each cannot be distinguished using toeprinting, as is now shown in Figure 4—figure supplement 2. Additionally, while the reviewer is correct that the use of a nonhydrolyzable GTP analog would normally be useful here, this is not the case with the IGR IRESs as these IRESs do not use or require GTP hydrolysis for 80S assembly. Also, the new version of Figure 4—figure supplement 1 shows far more robust 80S formation.

*6) “…the length mutants do not execute the first pretranslocation event…”. Although the toeprinting experiments using hygromycin B seem to support this conclusion (Figure 5), those done with the delta-1 and delta-2 mutants using cycloheximide tell the opposite (Figure 4). Please reconcile these results.*

The reviewer makes a valid point and we clearly did not explain this well. Examination of the toeprint gels shows visible toeprints at the pretranslocation position in the delta 1 and delta 2 mutants, even in the absence of the antibiotic. This is not the case in the G-rich and GGC mutants. This is consistent with the interpretation that the first translocation event is hindered (albeit to different degrees) in the length mutants. We have clarified this in the text.

Reviewer #2:*1) The work presented uses many different translation systems–yeast, shrimp,* E. coli*, rabbit retic and insect viruses. The need to utilize reagents from various origins is understandable and unavoidable due to inefficiencies of the current reconstituted systems, but a certain rigor is needed. I counted at least 4 mixed translation systems in this paper. Furthermore, tRNA can be from yeast or* E. coli*; similar true for the ribosomes that could be yeast or shrimp. It creates confusion, as sometimes it is impossible to deduce what combination of reagents has been used. It would be desirable if authors clearly indicate sources of ribosomes, factors and tRNA for every experiment.*

*While it is fully acceptable to use mixed model systems, a caution should be used when such systems are compared to each other. Curiously, Kieft et al. recently published a paper where it was demonstrated that translation initiation on CrPV IRES occurs via different mechanism in* E. coli*. The authors should show that initiation in the various mixed systems used in the current manuscript occurs via similar mechanisms. Perhaps a toeprint in each used system, as the paper already presents a toeprint for yeast ribosomes in RRL. It would be useful if authors would explain why they feel a need to use multiple systems, as it will help to understand how comparable those systems are. This is the biggest weakness of the current work, and one the authors should be able to readily address, given the robust mechanism of the IGR IRESs.*

Please refer to the response above.

*2) Dual luciferase assay data should be supported by experiments demonstrating reporter mRNA integrity and abundance. This is especially true when different reporter constructs are compared, and more so when changes are expected to affect mRNA structure. Northern blot of RRL samples used to measure luciferase activity would be highly preferable, if that is impossible authors should show the denaturing and native gel of the reporter mRNAs.*

Please refer to the response above.